# Tissue-specific activation of gene expression by the Synergistic Activation Mediator (SAM) CRISPRa system in mice

Charleen Hunt [1,2], Suzanne A. Hartford [1,2], Derek White[1], Evangelos Pefanis[1], Timothy Hanna[1], Clarissa Herman[1], Jarrell Wiley[1], Heather Brown[1], Qi Su[1], Yurong Xin[1], Dennis Voronin[1], Hien Nguyen[1], Judith Altarejos[1], Keith Crosby[1], Jeffery Haines[1], Sarah Cancelarich[1], Meghan Drummond[1], Sven Moller-Tank[1], Ryan Malpass[1], Jacqueline Buckley[1], Maria del Pilar Molina-Portela[1], Gustavo Droguett[1], David Frendewey[1], Eric Chiao[1], Brian Zambrowicz [1] & Guochun Gong [1✉]

CRISPR-based transcriptional activation is a powerful tool for functional gene interrogation; however, delivery difficulties have limited its applications in vivo. Here, we created a mouse model expressing all components of the CRISPR-Cas9 guide RNA-directed Synergistic Activation Mediator (SAM) from a single transcript that is capable of activating target genes in a tissue-specific manner. We optimized Lipid Nanoparticles and Adeno-Associated Virus guide RNA delivery approaches to achieve expression modulation of one or more genes in vivo. We utilized the SAM mouse model to generate a hypercholesteremia disease state that we could bidirectionally modulate with various guide RNAs. Additionally, we applied SAM to optimize gene expression in a humanized *Transthyretin* mouse model to recapitulate human expression levels. These results demonstrate that the SAM gene activation platform can facilitate in vivo research and drug discovery.

[1] Regeneron Pharmaceuticals, Inc., Tarrytown, NY, USA. [2] These authors contributed equally: Charleen Hunt, Suzanne A. Hartford. ✉email: guochun. gong@regeneron.com

Gene overexpression is a well-established approach to investigating biological pathways, screening for modifiers, and modeling disease. Once limited to yeast, such studies can now be pursued in mouse embryos and mouse embryonic stem cells (mESC) using random insertion of nucleic acids. Both viral and non-viral approaches are possible, but the genetic lesions these techniques produce can be difficult to propagate due to low germline transmission rates[1,2]. As transgenic over-expression is usually achieved by random integration of multiple copies of an expression vector, the result is unpredictable and difficult to reproduce. Additionally, the transgene is often a highly engineered construct with an artificial transcriptional promoter and signaling elements that do not reproduce the gene's natural expression[3,4]. Site-specific integrases can direct integration at a desired site, but they require generation of an acceptor allele prior to the targeting event[5–7]. Hydrodynamic delivery (HDD) of plasmid DNA is a common method for overexpression in the liver but does not achieve stable expression and is associated with transient tissue damage and toxicity[8].

The development of programable nucleases, such as zinc finger nucleases and CRISPR-Cas9, has revolutionized genome modulation studies. Gene activation by engineered zinc finger nucleases is possible but hindered by challenges in design and production of sequence-specific zinc finger proteins[9,10]. Conversely, the plasticity inherent to CRISPR-Cas9 allows virtually any type of genetic lesion to be accurately produced and transmitted through the germline. Recently, Cas9 variants have been adapted for gene expression modulation. Inactivating the catalytic domains of Cas9 converts the enzyme into a nuclease-deficient (dCas9) variant that binds, but does not cut, target DNA in a guide RNA (gRNA)-dependent manner. Coupling transcriptional activation modules to dCas9 can create a CRISPRa system that can be directed to activate a gene of interest[11–14].

Current strategies for achieving in vivo activation of gene expression suffer from serious technological limitations. Adeno-Associated Virus (AAV)-based vectors are constrained to a 4.7 kb cargo making it difficult to incorporate the necessary elements of a CRISPRa system. This can be somewhat mitigated with the use of active Cas9 directed to its target by truncated "dead gRNAs" that support low-level cutting[15]. These gRNAs can be introduced along with transcription factors by AAV, but the presence of low-level cutting may complicate findings[15,16]. Small cas9 variants can function as nuclease-deficient activators, but are less efficient than full versions and may still require two AAV vectors[13,17–19].

We explored whether a CRISPR-based transcriptional activation (CRISPRa) system targeted to a specific locus in the mouse genome would address the limitations of existing technologies while also providing the advantage of consistent expression across generations of mice. Although such activation systems have been developed and chimeric mice produced, a systematic in vivo characterization of these alleles has not been described[16,20]. An in vitro comparison of various CRISPRa platforms identified the SAM system as the most consistent activation approach, making it an ideal platform to evaluate in vivo[21]. In vitro, the SAM complex is expressed from three lentiviruses encoding dCas9, transcriptional activators, and gRNAs with an MS2 aptamer tracr variant (tracr$^{MS2}$)[11,15]. This three-component system allows for flexibility in cell culture but is less desirable in vivo where transgenic insertions of three components may integrate at different frequencies, be silenced over time, or segregate independently during breeding. To overcome these difficulties, we created a single transcript driven by the Gt(ROSA)26Sor (R26) promoter for coding dCas9 nuclease, transcription factors, and MCP[22,23]. Using several modes of gRNA delivery, we demonstrate the versatility of SAM mice for both global and tissue-specific gene induction with utility for disease modeling. We show that multiple genes can be upregulated via a gRNA array and that activation can be restricted to the target gene. Hence, this SAM system combines gene expression modulation with the consistency of an in vivo model to support a flexible approach to accelerating basic research.

## Results

**Characterization and functional validation of R26$^{SAM}$ allele.** We generated a targeted allele in mESCs that comprises of Cre-dependent dCas9-SAM expression from a single open reading frame[24]. The dCas9-SAM allele contains two hybrid fusion proteins: dCas9-VP64 (dCas9 fused to VP64 transcriptional activator) and MCP-HSF1-P65 (MCP, the coat protein from MS2 RNA bacteriophage fused to transcriptional activators HSF1 and P65 linked by a 2A peptide[25–27]. The MCP fused to transcription factors binds the MS2 aptamers to bring VP64, HSF1, and P65 together to synergistically activate targets. We inserted a loxP-stop-loxP (LSL) stop cassette upstream of the SAM coding sequences to block expression from the R26 promoter (Fig. 1a). Western blot analysis demonstrated dCas9 expression in mESC clones only following removal of the stop cassette by Cre recombinase (Fig. 1b).

The activation efficiency of this CRISPRa allele was tested by targeting six genes that have low or no expression in mESCs, as well as tissue-specific expression in mice[28–30]. The integrated gRNA arrays were composed of tandem tracr$^{MS2}$ gRNA expression units, each driven by a U6 promoter (Fig. 1c). These arrays were targeted to the second R26 allele of SAM mESCs (R26#a[Gene], where [#]a[Gene] indicates the number of activation gRNAs against the specified gene).With the exception of gRNAs targeting Tsx, protospacers (target-specific regions of the gRNA) were selected using a window of 200–300 bases upstream of the predicted transcriptional start site (TSS) (Fig. 1d)[11,31,32]. The design of R26$^{4aTsx}$ was unique in that two gRNAs were designed proximal to the TSS while the other two were designed distally to interact with an annotated enhancer region (Supplementary Fig. 1a)[33–35]. We used RNA-Seq to establish the basal expression of each target gene (number of mRNA transcripts per million Next Generation Sequence (NGS, Illumina) reads, TPM) in R26$^{SAM/+}$ cells (Supplementary Fig. 1b, black bars). We then performed quantitative reverse transcription PCR (RT-qPCR) on eight unique clones from each gRNA array targeted line to assay the specific activation of the target genes relative to their expression in R26$^{SAM/+}$ cells (Supplementary Fig. 1b, blue bars). As previously described, the lower the basal expression (black bars), the higher the degree of activation achieved (blue bars)[11,31,36]. Specifically, the strongest activations were associated with targets that do not have detectable transcripts in R26$^{SAM/+}$ cells (Ttr, Celrr, Rs1 and Alb).

*There is a strong correlation in gene expression between R26$^{SAM}$ and R26$^{LSL-SAM}$ mESCs.* A Pearson's correlation coefficient (r-value) comparison of the R26$^{LSL-SAM}$ and R26$^{SAM}$ transcriptomes confirmed that SAM expression without gRNAs did not significantly impact global RNA expression (R26$^{LSL-SAM}$ vs wildtype (WT or R26$^{+/+}$): r = 0.994, R26$^{SAM}$ vs R26$^{LSL-SAM}$: r = 0.998, Supplementary Fig. 1c). To determine the global effects of SAM-mediated gene activation from an integrated array, transcriptomes of four R26#a[Gene] lines were compared to R26$^{SAM/+}$ via RNA-seq. Expression of all target genes in R26#a[Gene] lines was substantially upregulated compared to R26$^{SAM/+}$ (Fig. 1e–h, blue stars). After disregarding noise from low-expressed genes (<0.1 TPM), the r-value of gene expression between each R26$^{SAM/3a[Gene]}$ and R26$^{SAM/+}$ was >0.99 in all cases indicating that the mESC transcriptome is minimally perturbed in the evaluated

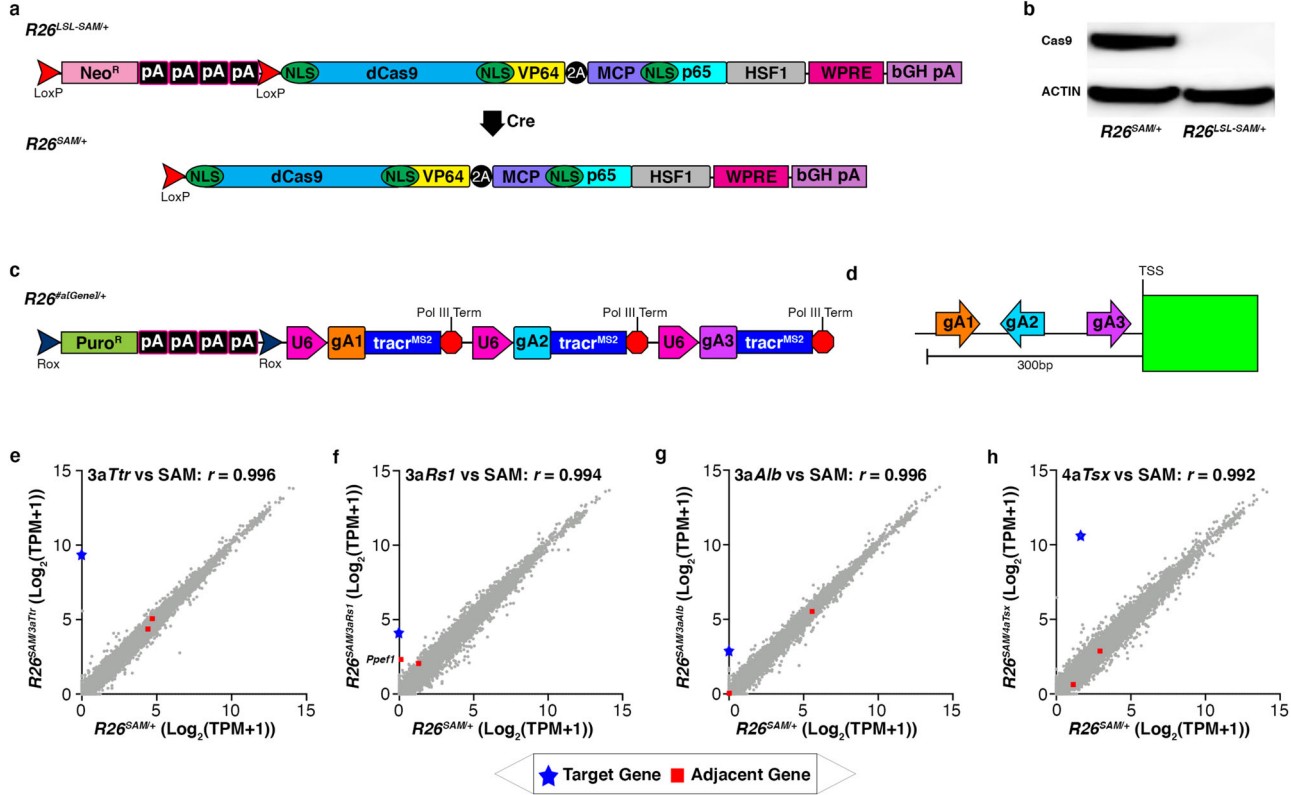

**Fig. 1 Characterization and validation of $R26^{SAM/+}$ in mESCs. a** Schematic of $R26^{LSL-SAM/+}$ and $R26^{SAM/+}$. dCas9 is fused to VP64 with a nuclear localization sequence (NLS). MCP, P65, HSF1 are fused together with an intervening NLS sequence. The hybrid protein sequences are linked via a P2A peptide. Upon treatment with Cre recombinase, the loxP-stop-loxP cassette (LSL) is excised. **b** A representative western blot showing dCas9$^{SAM}$ (219 kDA) expression in targeted mESCs. dCas9$^{SAM}$ signals are normalized to actin (41 kDa). Three independent repeats have been completed with similar results. **c** Generalized schematic of $R26$ targeted guide arrays where "#a" indicates the number of activating guides included in the array and [Gene] refers to the target gene. Each guide is driven by a U6 promoter and separated by an extended RNA Pol III termination sequence. **d** Illustration of general SAM guide design approach. Guides are selected within 300 bases of the transcriptional start site (TSS) of each gene. **e–h** RNA-seq characterization of mESC lines harboring targeted gRNA arrays versus the parental $R26^{SAM/+}$ line. The transcriptome of each clone was sequenced with five technical replicates to provide the number of target-specific transcripts per million (TPM) sequence reads. The target gene is noted with a blue star and the adjacent genes with red squares. Statistics: Pearson's correlation coefficient was utilized to determine the r-value (**e**) r = 0.994, (**f**) r = 0.996, (**g**) r = 0.992, (**h**) r = 0.996.

activation lines. However, in one instance, activation of an adjacent gene was observed: *Ppef1* was upregulated nearly 45-fold over $R26^{SAM/+}$ expression in $R26^{SAM/3aRs1}$ cells (Fig. 1f, annotated red square). Mapping of *Ppef1* revealed that it is 47-kilobases upstream of *Rs1* in a head-to-head orientation[33,37]. RNA-seq data was further analyzed for the expression of any gene within 500-kilobases of a target gene and slight deviations were noted only with regard to genes neighboring *Ttr* and *Tsx*. RT-qPCR evaluation of the same clones confirmed a 10-fold increase in *Ppef1* expression in $R26^{SAM/3aRs1}$ cells and non-significant differences in neighboring gene expression in $R26^{SAM/3aTtr}$, $R26^{SAM/3aTsx}$, and $R26^{SAM/3aAlb}$ (Supplementary Fig. 1a, d–g).

**Characterization of SAM activity in mice.** Expression of SAM was observed via western blot across twelve tissues collected from $R26^{SAM/+}$ mice, whereas no expression was detected in WT controls (Supplementary Fig. 2a). RT-qPCR confirmed minimal differences in SAM mRNA detected in tissues (Supplementary Fig. 2b). These data confirm that the all-in-one SAM mRNA can be expressed from the $R26$ locus and can produce abundant protein in tissues to support gene activation studies in vivo.

Characterization of the activation capacity of CRISPRa in SAM mice was focused on the $R26^{SAM/3aTtr}$ allele due to the ease of measuring circulating TTR for activation levels. Circulating serum TTR was quantified monthly through enzyme-linked

immunosorbent assay (ELISA) in male $R26^{SAM/3aTtr}$ mice and controls. $R26^{SAM/3aTtr}$ mice maintained circulating TTR at a level that was 2.5-fold higher than $R26^{SAM/+}$ or WT mice (Fig. 2a). In a separate study, we investigated if the integrated array was capable of turning on *Ttr* expression in all tissues and indeed found elevated *Ttr* expression in the assayed tissues as compared with the $R26^{SAM/+}$ control (Fig. 2b and Supplementary Fig. 2c). Mirroring prior observations, tissues with higher endogenous expression of the target gene achieved lower upregulation (for example, liver expression was upregulated 2.5-fold), whereas tissues with low or no endogenous expression, achieved higher fold changes (as in the lung with 98,000-fold upregulation) as compared to $R26^{SAM/+}$ mice (Supplementary Fig. 2d). Next, we confirmed global CRISPRa activity for a second gRNA array targeting retina specific *Rs1*. $R26^{SAM/3aRs1}$ mice were generated with *Rs1* expression elevated in all tissues, including a 241,000-fold increase in the liver (Fig. 2c and Supplementary Fig. 2e). *Ppef1*, but not *Cdkl5*, was activated in tissues, consistent with the observation in mESCs (Fig. 1f and Supplementary Fig. 2f–i).

A comparison of tissues taken from $R26^{+/+}$, $R26^{SAM/+}$, and $R26^{SAM/3aTtr}$ mouse lines highlighted significant correlation ($r \geq 0.99$) regardless of the expression of SAM or arrayed gRNAs (Fig. 2d and Supplementary 3a, b). Evaluation of RNA-seq and RT-qPCR data from $R26^{SAM/3aTtr}$ tissues also confirms that neighboring genes were not significantly perturbed (Supplementary Fig. 3c).

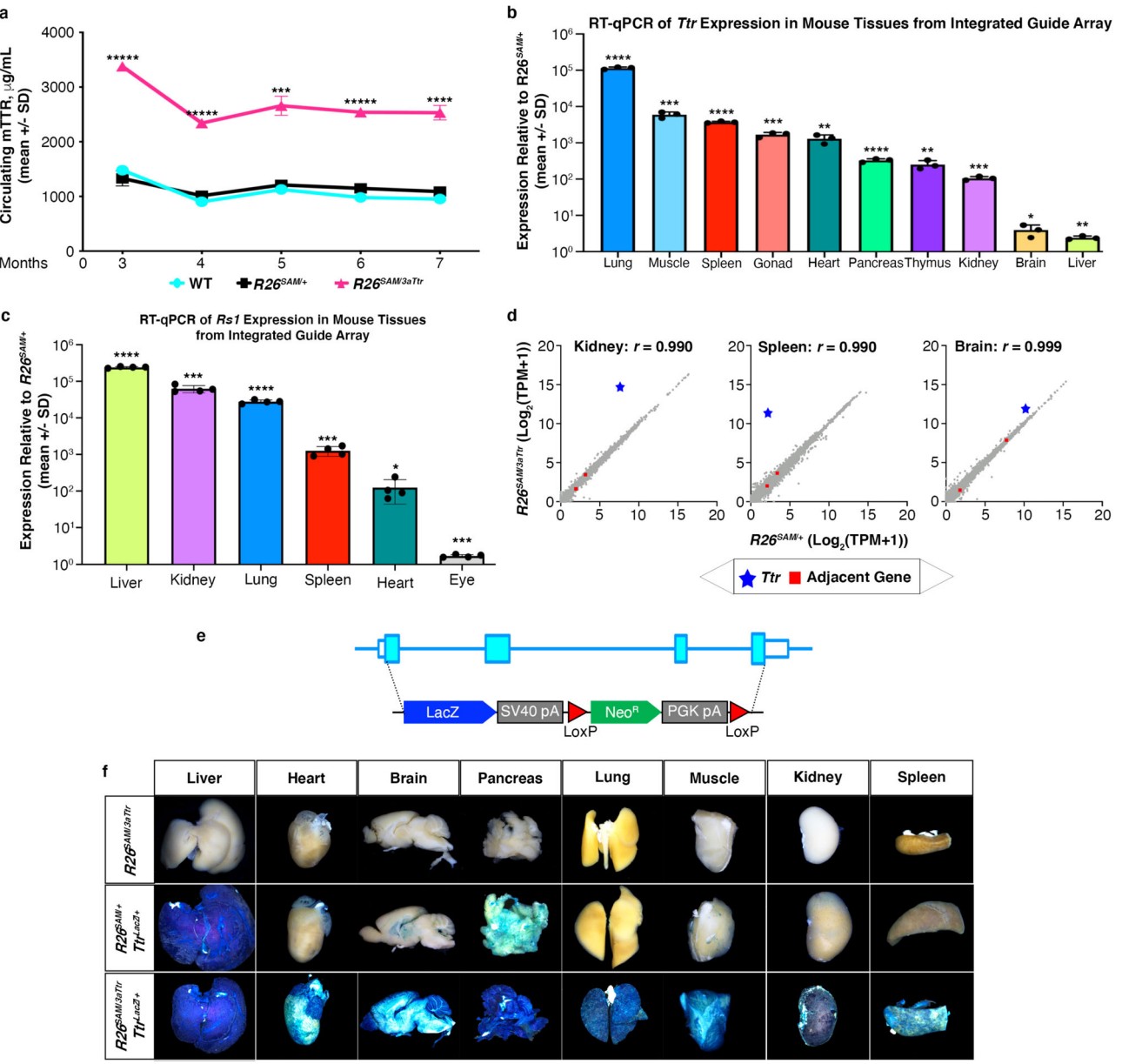

**Fig. 2 Validation of $R26^{SAM}$ in mice. a** Mouse serum TTR levels were evaluated in $R26^{SAM/3aTtr}$ ($n = 7$), $R26^{SAM/+}$ ($n = 4$) and WT ($n = 4$) for 7 months with monthly bleeds starting at 3 months. ELISA was used to determine protein levels in each mouse and values were plotted as the mean per group +/− SD. Monthly $p$-values can be found in Supplemental Table 2. Two independent replicates were conducted. **b** RT-qPCR determination of $Ttr$ expression levels per tissue in $R26^{SAM/3aTtr}$ mice ($n = 3$) relative to $R26^{SAM/+}$ plotted as the mean per group +/− SD. $p$-values of each tissue can be found in Supplemental Table 2. **c** RT-qPCR analysis of $Rs1$ expression in $R26^{SAM/3aRs1}$ mouse tissues ($n = 4$ mice per tissue) plotted as the mean per group +/− SD. $p$-values of each tissue can be found in Supplemental Table 2. **d** RNA-seq characterization of $R26^{SAM/3aTtr}$ mouse tissues versus the parental $R26^{SAM/+}$ line. The transcriptome of each mouse tissue from five mice were sequenced to determine the number of target-specific TPM. The target gene is noted with a blue star and the adjacent genes with red squares. Kidney: $r = 0.990$, Spleen: $r = 0.990$, Brain: $r = 0.999$. **e** Schematic of LacZ knockout approach. The $Ttr$ mouse coding sequence was replaced in mESCs starting at ATG with the LacZ coding sequence ($Ttr^{LacZ/+}$). **f** Tissues harvested from $R26^{SAM/3aTtr}$; $R26^{SAM/+}$ mice ($n = 3$), $R26^{SAM/+}$; $Ttr^{LacZ/+}$ mice ($n = 3$) and $R26^{SAM/3aTtr}$;$Ttr^{LacZ/+}$ mice ($n = 3$) stained with X-gal. Statistics: Asterisks (*) indicates significance, and the number of asterisks (*) indicates the number of 0s after the decimal point. One-tailed, unpaired Student's $t$-test for (**a–c**) and Pearson's correlation for (**d**).

A visual readout of $Ttr$ upregulation was achieved using female $R26^{SAM/3aTtr}$ mice in which the coding sequence of the $Ttr$ gene was replaced with the *Escherichia coli* lacZ gene encoding beta-galactosidase (Fig. 2e)[38]. Mice containing $R26^{SAM/3aTtr}$;$Ttr^{LacZ/+}$ showed strong beta-galactosidase activity. Notably, tissues with low or no expression of $Ttr$ under normal circumstances, such as spleen, now showed lacZ staining (Fig. 2f). The induced expression profile of the reporter allele indicates that SAM

activation is independent of the normal expression profile of the gene.

**Gene activation by non-transgenic tracr^MS2 gRNA delivery methods in SAM mice.** AAV is an attractive gene delivery approach due to its persistence in non-dividing cells, mild pathogenicity, defined tropisms, and proven therapeutic potential[39–41]. We generated AAV8 particles to express various

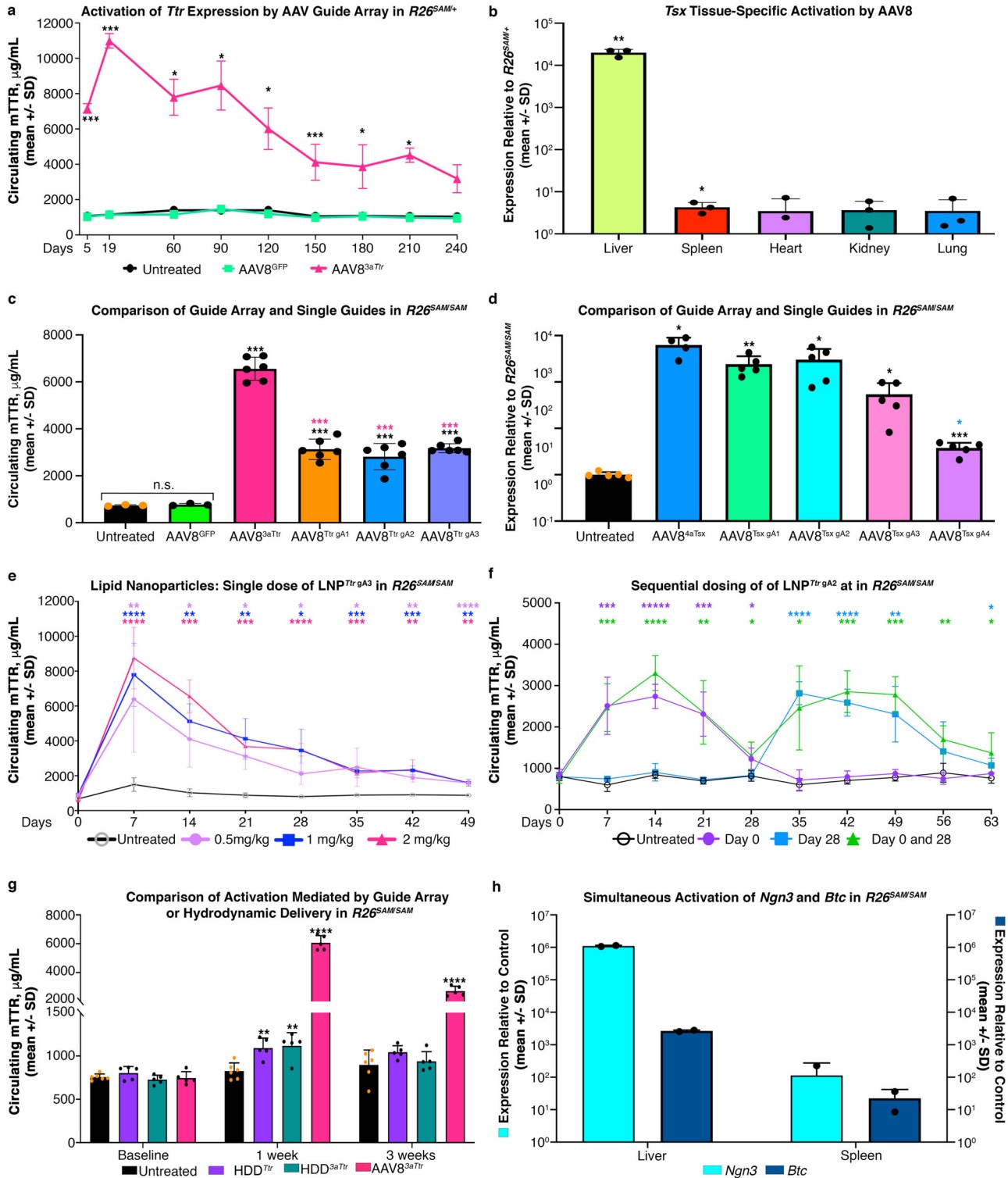

gRNA arrays; Sanger sequencing of these AAV genome constructs confirmed the stability of arrayed guides (Supplementary Fig. 4a, b and Supplementary Tables 8-27). The degree and duration of gene upregulation from a single dose of AAV8[3aTtr] was evaluated in male mice by monitoring circulating TTR. Post-delivery of AAV8[3aTtr], $R26^{SAM/+}$ mice exhibited a sevenfold increase in circulating TTR while untreated animals and animals infected with a control virus containing GFP (AAV8[GFP]) had baseline levels of TTR that did not alter over the course of the study (Fig. 3a). By day 19 post infection, serum TTR levels had

increased 11-fold over controls. Liver turnover is low with a typical hepatocyte life span of 200 to 400 days[42–44]. The AAV does not have replication machinery so it is not surprising that the TTR values gradually decreased over the next eight months without reaching normal levels.

Next, experiments were performed targeting $Tsx$ to determine if upregulation of loci with low endogenous expression levels could be achieved in $R26^{SAM/+}$ mice. Three weeks after injection of AAV8[4aTsx] in male SAM mice, five tissues were harvested for RT-qPCR. $Tsx$ expression, which is not typically found in the

**Fig. 3 SAM guide delivery approaches. a** AAV8$^{3aTtr}$ ($n = 3$) and AAV8$^{GFP}$ ($n = 2$) were delivered to $R26^{SAM/+}$ mice. ELISA was employed to determine protein levels along with untreated $R26^{SAM/+}$ ($n = 2$). Values were plotted as the mean per group $+/-$ SD. Monthly $p$-values can be found in Supplementary Table 3. Two independent studies were conducted with similar results. **b** AAV8$^{4aTsx}$ was delivered to $R26^{SAM/+}$ mice ($n = 3$) to evaluate the tissue-specific induction of $Tsx$. Expression values relative to $R26^{SAM/+}$ were plotted as the mean per group $+/-$ SD. Liver: $p = 0.001026$; Spleen: $p = 0.01138$. **c** Protein induction by arrayed guide viral delivery to $R26^{SAM/SAM}$ mice was compared to single guide viral delivery ($n = 3$ for controls, $n = 6$ for treated). Serum levels were plotted as the mean per group $+/-$ SD. $p$-values of each array comparison can be found in Supplementary Table 3. **d** AAV8$^{4aTsx}$ comprises two guides targeting the 300 bp directly upstream of the TSS and another two guides targeting an upstream enhancer region (750 bp upstream). qPCR was completed on livers harvested from $R26^{SAM/SAM}$ treated with either AAV delivered guide array (4a$Tsx$) or individual guides from the array ($Tsx$ gA1-4) ($n = 5$). Expression values relative to $R26^{SAM/+}$ were plotted as the mean per group $+/-$ SD. $p$-values of each array comparison can be found in Supplemental Table 3. **e** LNP particles were formulated with synthetic $Ttr$ gA3 SAM guides and introduced to $R26^{SAM/SAM}$ mice ($n = 3$). Protein expression levels were determined by ELISA with weekly bleeds. Daily $p$-values can be found in Supplementary Table 3. All values are plotted as mean $+/-$ SD. Two independent replicates were conducted with similar results. **f** LNP particles were formulated with 0.5 mpk of synthetic $Ttr$ gA2 SAM guide and introduced to $R26^{SAM/SAM}$ mice ($n = 5$) at zero weeks and/or 4 weeks. Protein expression levels were determined by ELISA with weekly bleeds. Daily $p$-values can be found in Supplementary Table 3. All values are plotted as mean $+/-$ SD. **g** Overexpression by HDD delivery of $Ttr$ cDNA or a plasmid expressing arrayed SAM guides was compared to AAV8$^{3aTtr}$ in $R26^{SAM/SAM}$ mice ($n = 5$, per group). TTR levels were determined by ELISA and plotted as mean per group $+/-$ SD. Untreated vs Array week 1: $p = 0.000001$; Untreated vs. Array week 2: $p = 0.000001$; Untreated vs HDD$^{3aTtr}$ week 1: $p = 0.003585$; Untreated vs. HDD$^{Ttr}$ plasmid week 1: $p = 0.002222$. Two independent replicates were conducted with similar results. **h** A guide array expressing two activating guides each to $Ngn3$ and $Btc$ was delivered to $R26^{SAM/SAM}$ mice ($n = 2$). Five tissues were collected and expression relative to $R26^{SAM/SAM}$ was plotted as mean per group $+/-$ SD. Statistics: Asterisks (*) indicates significance, and the number of asterisks indicates the number of 0s after the decimal point. One-tailed, unpaired Student's $t$-test for (**a**-**j**).

---

liver, exhibited a 20,000-fold increase over control mice. $Tsx$ expression in potential AAV8 off-target tissues, including spleen and lung, was mildly perturbed (Fig. 3b).

The resulting viral induction of $Ttr$ in $R26^{SAM/+}$ mice prompted us to investigate if working in a $R26^{SAM/SAM}$ background may increase activation further. RT-qPCR was completed on five tissues harvested from SAM mice to characterize differences in transcript levels between heterozygous and homozygous mice. In all tissues, there was no more than one threshold cycle (Ct) difference between genotypes indicating a minimal difference in expression (Supplementary Fig. 5a). To validate activity, AAV8$^{3aTtr}$ was injected and serum TTR levels were used to infer dCas9 SAM function. No significant difference was observed between $R26^{SAM/+}$ and $R26^{SAM/SAM}$ mice (Supplementary Fig. 5b). In addition to genotype, models may vary between male and female study cohorts. We used the same approach as above to verify that $R26^{SAM/SAM}$ alleles function similarly regardless of the cohort gender (Supplementary Fig. 5c).

It was not clear whether the enhanced gene expression observed with AAV8$^{3aTtr}$ and AAV8$^{4aTsx}$ viruses was due to one highly active gRNA or to synergistic activity from the entire array. To distinguish these two scenarios, AAV8 particles expressing each of the individual gRNAs in $3aTtr$ (e.g., AAV8$^{Ttr-gA1}$, AAV8$^{Ttr-gA2}$, or AAV8$^{Ttr-gA3}$, Supplementary Fig. 4c) and 4a$Tsx$ were delivered to female $R26^{SAM/SAM}$ mice. Individual $Ttr$ gRNAs each resulted in a threefold increase in circulating TTR while the array mediated a sevenfold increase (Fig. 3c). RT-qPCR performed on harvested tissues confirmed that a similar titer of each virus was administered to the study cohort (Supplementary Fig. 4d). $Tsx$ expression was examined by RT-qPCR analysis on livers harvested from male $R26^{SAM/SAM}$ mice injected with either AAV8 delivered gRNA array or individual gRNAs. Despite being ~750 bases upstream, the distal gRNAs ($Tsx$-gA2 and gA3) promoted activation similar to, or better than, proximal gRNAs $Tsx$-gA1 and gA4 (Fig. 3d and Supplementary Fig. 1a). However, the four-gRNA array activated $Tsx$ expression above any single gRNA virus activation.

To ascertain whether other gRNA delivery methods could afford the same success as AAV, we injected male $R26^{SAM/SAM}$ mice with LNPs containing gRNA $Ttr$-gA3 (LNP$^{Ttr-gA3}$) at three doses: 0.5 milligrams per kilogram of mouse body weight (mpk), 1 mpk, and 2 mpk. This transient delivery method produced dose-dependent gene activation for ~3 weeks and elevated serum

TTR levels for more than a month (Fig. 3e). The lowest dose yielded a sevenfold increase while the highest dose yielded a 15-fold increase. To our knowledge, this is the first instance of gene activation by LNP delivery of tracr$^{MS2}$ gRNA to a mouse model. A second study was initiated to evaluate the impact of sequential dosing. All mice were injected with 0.5 mpk of LNP formulated with $Ttr$-gA2 at the start of the study and blood draws were taken weekly. Subsets of these mice were dosed with another 0.5 mpk LNP at 2-weeks or 4-weeks, with additional naive mice injected to confirm LNP function (Fig. 3f and Supplementary Fig. 5d). In all cases, redosing successfully boosted $Ttr$ expression.

HDD is a technique to overexpress genes in mouse livers. To test HDD as a modality for SAM activation, we compared hydrodynamic injection of male mice with a plasmid expressing the $Ttr$ gRNA array (HDD$^{3aTtr}$; Supplementary Fig. 4e) to viral injection with AAV8$^{3aTtr}$. For reference, we performed HDD on SAM mice with a conventional HDD plasmid expressing $Ttr$ cDNA from the human $Ubiquitin$ promoter (HDD$^{Ttr}$, Supplementary Fig. 4f). Serum collected 3 weeks post treatment showed that AAV8$^{3aTtr}$ elicited a stronger response than either standard HDD$^{3aTtr}$ or HDD$^{Ttr}$ (Fig. 3g). HDD$^{Ttr}$ reached its maximum circulating TTR level at 1-week post injection with an average 1.3-fold increase across the study mice. HDD$^{3aTtr}$ performed similarly with an average 1.5-fold increase. The five mice treated with AAV8$^{3aTtr}$ yielded an average sevenfold increase in serum TTR.

Some biological processes require simultaneous overexpression of two or more genes to induce a phenotype. The efficiency of viral delivery and the observation that single gRNAs can impact gene expression suggest the possibility to activate expression of more than one gene with a single viral construct. We endeavored to activate two genes, $Ngn3$ and $Btc$, with a single AAV8 harboring four-gRNA expression units, two gRNAs per target. Three weeks after delivery of AAV8$^{2aNgn3-2aBtc}$ to female $R26^{SAM/SAM}$ mice, tissues were collected and activation was determined by RT-qPCR (Fig. 3h). $Ngn3$ achieved greater than 1,000,000-fold increase in relative expression in the liver while $Btc$ demonstrated a 3000-fold increase. In both cases, expression in the spleen was also induced, albeit to a lesser extent than the liver.

**Timing and tissue-specific gene upregulation by controlled SAM activation.** The dependence of SAM expression on removal of the LSL in $R26^{LSL-SAM/+}$ mice should enable spatial and

temporal control of gene activation. To test this hypothesis, male $R26^{LSL-SAM/+}$ and $R26^{SAM/+}$ mice were treated with LNP carrying Cre mRNA ($LNP^{Cre}$) with and without $AAV8^{3aTtr}$ injection (Fig. 4a). No increase in TTR was seen in $R26^{LSL-SAM/+}$ mice treated with $AAV8^{3aTtr}$ or $LNP^{Cre}$ alone. In contrast, combined delivery of $LNP^{Cre}$ and $AAV8^{3aTtr}$ to $R26^{LSL-SAM/+}$ increased circulating TTR sevenfold, roughly equivalent to that observed in $R26^{SAM/+}$ treated with $AAV8^{3aTtr}$ alone. Western blot analysis confirmed the induction of dCas9 protein in $R26^{LSL-SAM}$ livers, but not in kidney and spleen, after $LNP^{Cre}$ delivery (Fig. 4b). To combine temporal SAM expression with tissue specificity, male $R26^{LSL-SAM/4aTsx}$ mice were created. Although $Tsx$ activating gRNAs were ubiquitously expressed, there was no increase in $Tsx$ expression until treatment with $LNP^{Cre}$ (Fig. 4c). These data show the $R26^{LSL-SAM}$ animals allow both tissue-specific and temporal regulation of gene activation.

Some tissues are hard to target due to lack of tools with the appropriate tropism. Although central nervous system (CNS)-specific AAV serotypes have been identified, there is still difficulty with transduction efficiency in some mouse strains, as well as a need for more selective transduction of CNS cell types[19,45–47]. With a goal of targeting the arcuate nucleus of the hypothalamus, we performed stereotactic delivery of $AAV8^{3aTtr}$ to male mice (Supplementary Fig. 5e). RNAscope analysis on the resulting brain sections revealed strong staining for $Ttr$ mRNA that colocalized with staining for $Pomc$ mRNA, a gene specifically active in the arcuate nucleus[48]. In contrast, WT mice injected with $AAV8^{3aTtr}$ show $Pomc$ expression without $Ttr$ (Fig. 4d, e). Additional stereotactic surgeries yielded $Ttr$ overexpression in the dentate gyrus further confirming that the SAM system can be implemented surgically (Supplementary Fig. 5f).

To support the study of tissue regions that cannot easily be accessed by surgery or viral delivery, we interrogated the SAM mouse for use in explant studies. The organ of Corti is a specialized region within the cochlea responsible for transduction of auditory signals[49]. The cochlea is difficult to access surgically in a live animal without causing artifacts due to tissue damage and cell line models do not exist. However, the neonatal murine organ of Corti can be removed intact for ex vivo organ culture[50,51]. The organ of Corti from $R26^{SAM/SAM}$ neonates was harvested and treated with $AAV8^{3aTtr}$, $AAV8^{3aPcsk9}$, or $AAV8^{3aLdlr}$ (Fig. 4f). $Ttr$ and $Ldlr$ expression levels were elevated by an average of 20-fold compared to WT controls and $Pcsk9$ was increased 275-fold (Fig. 4g). These data demonstrate that SAM mouse tissues can be used as a tool for gene activation in explant models.

### Application of SAM facilitates generation of disease models.
The observed versatility of expression modulation in SAM mice encouraged the evaluation of disease modeling potential. We chose to focus on $Pcsk9$ as it plays a major regulatory role in cholesterol homeostasis by reducing LDLR levels on the plasma membrane[52]. Upon internalization, PCSK9 bound LDLR is unable to escape digestion by the lysosome and, consequently, cannot be recycled back to the cell surface. $Pcsk9$ overexpression leads to increased degradation of LDLR and, inevitably, increased circulating LDL (Supplementary Fig. 6a)[52,53]. Conversely, an increase in $Ldlr$ expression leads to more LDLR on the cell surface, which translates to lower circulating LDL (Supplementary Fig. 7a)[52,53].

Hypercholesteremia, (total cholesterol >240 mg/dL) is estimated to affect one in every 250–500 people and represents a major contribution to the worldwide burden of premature cardiovascular disease[54]. We utilized male $R26^{SAM/SAM}$ mice to overexpress $Pcsk9$ and drive mouse total cholesterol levels close to

300 mg/dL, well in range of human hypercholesteremia. Twelve to 15-week-old mice were fasted overnight before determining their baseline body weight and serum chemistry profile. Groups of five mice were injected with one of four viruses: $AAV8^{3aPcsk9}$, $AAV8^{Pcsk9-gA1}$, $AAV8^{Pcsk9-gA2}$, or $AAV8^{Pcsk9-gA3}$ and blood was drawn bi-weekly for liver lipid profiles. Mice treated with $AAV8^{Pcsk9-gA1}$ reached greater than eightfold elevated LDL-cholesterol by the week two bleed (Fig. 5a and Supplementary 6b). $AAV8^{3aPcsk9}$ reached its maximum LDL level 2-weeks later with an average of sixfold over baseline. $AAV8^{Pcsk9-gA2}$ and $AAV8^{Pcsk9-gA3}$ each resulted in modest twofold increases of LDL. A similar trend was observed in total cholesterol levels, with $AAV8^{3aPcsk9}$ and $AAV8^{Pcsk9-gA1}$ resulting in the highest cholesterol levels (Fig. 5b). Contrasting our previous results with $Ttr$, $Tsx$, and $Rs1$, a single gRNA outperformed the $Pcsk9$ array. It is possible the poor performance of $AAV8^{Pcsk9-gA2}$ and $AAV8^{Pcsk9-gA3}$ inhibited the performance of gA1 in the array. With respect to high-density lipoprotein (HDL) and triglycerides, no remarkable differences were detected between injected and control mice (Fig. 5c and Supplemetary 6c). RT-qPCR of livers after the final timepoint confirm that the gA1 binding site is the main driver of $Pcsk9$ overexpression (Supplementary Fig. 6d). The same approach was applied to female SAM mice and a similar trend was observed (Fig. 5d–f and Supplementary 6e). ELISA quantification of circulating PCSK9 highlighted that serum levels were substantially elevated in mice treated with $AAV8^{3aPcsk9}$ and $AAV8^{Pcsk9-gA1}$ (Supplementary Fig. 6f).

An increase in $Ldlr$ expression should lead to a reduction in circulating LDL-cholesterol over time. Mice typically have a low-baseline LDL-cholesterol level (averaging 6–10 mg/dL) and so we placed study mice on a high-fat diet (HFD) for the duration of the study to elevate cholesterol levels prior to induction of $Ldlr$. We evaluated viral delivery of single gRNAs or a gRNA array targeting $Ldlr$ in female $R26^{SAM/SAM}$ mice. Baseline liver lipid profiles were determined before mice were placed on HFD with blood draws every 2 weeks. After 6 weeks on HFD, groups were injected with the 3-guide AAV array or an individual guide. Blood draws continued bi-weekly until the final 12-week study endpoint. $AAV8^{Ldlr-gA1}$ facilitated a 5.6-fold reduction in LDL, $AAV8^{Ldlr-gA2}$ achieved a 3.4-fold reduction while $AAV8^{Ldlr-gA3}$ had a 2.2-fold elevation in LDL. When all gRNAs were included in $AAV8^{3aLdlr}$, a 1.4-fold decrease was observed (Supplementary Fig. 7b, c). In contrast to the $Pcsk9$ study, modulation of $Ldlr$ expression resulted in a reduction of circulating HDL and total cholesterol (Supplementary Fig. 7d, e). No clear trend was observed in triglyceride levels (Supplementary Fig. 7f). The relatively low performance of the array may be attributed to the ineffective gRNA (gA3) as supported by RT-qPCR (Supplementary Fig. 7g).

To elucidate if a modified array including only the active guides would facilitate a more robust reduction of LDL, we generated an AAV8 array expressing $Ldlr$-gA1 and gA2 ($AAV8^{2aLdlr}$). $AAV8^{2aLdlr}$ was delivered to female $R26^{SAM/SAM}$ mice who were maintained on a HFD in a side-by-side comparison to the original AAV8 guide sets as above. As before, $AAV8^{Ldlr-gA1}$ and $AAV8^{Ldlr-gA2}$ outperformed $AAV8^{Ldlr-gA3}$. The three-guide array facilitated a 1.7-fold decrease in LDL; however, the two-guide array mediated a ninefold reduction in circulating LDL (Fig. 5g and Supplementary Fig. 7h). These data demonstrate that the inclusion of an inefficient guide in a delivered array may negatively impact the activation of the intended target. This is further illustrated by the relative expressions of $Ldlr$ at the final endpoint: 2a$Ldlr$ managed a fourfold increase in expression whereas 3a$Ldlr$ is comparable to the untreated controls (Supplementary Fig. 7i). Furthermore, $AAV8^{2aLdlr}$ facilitated a

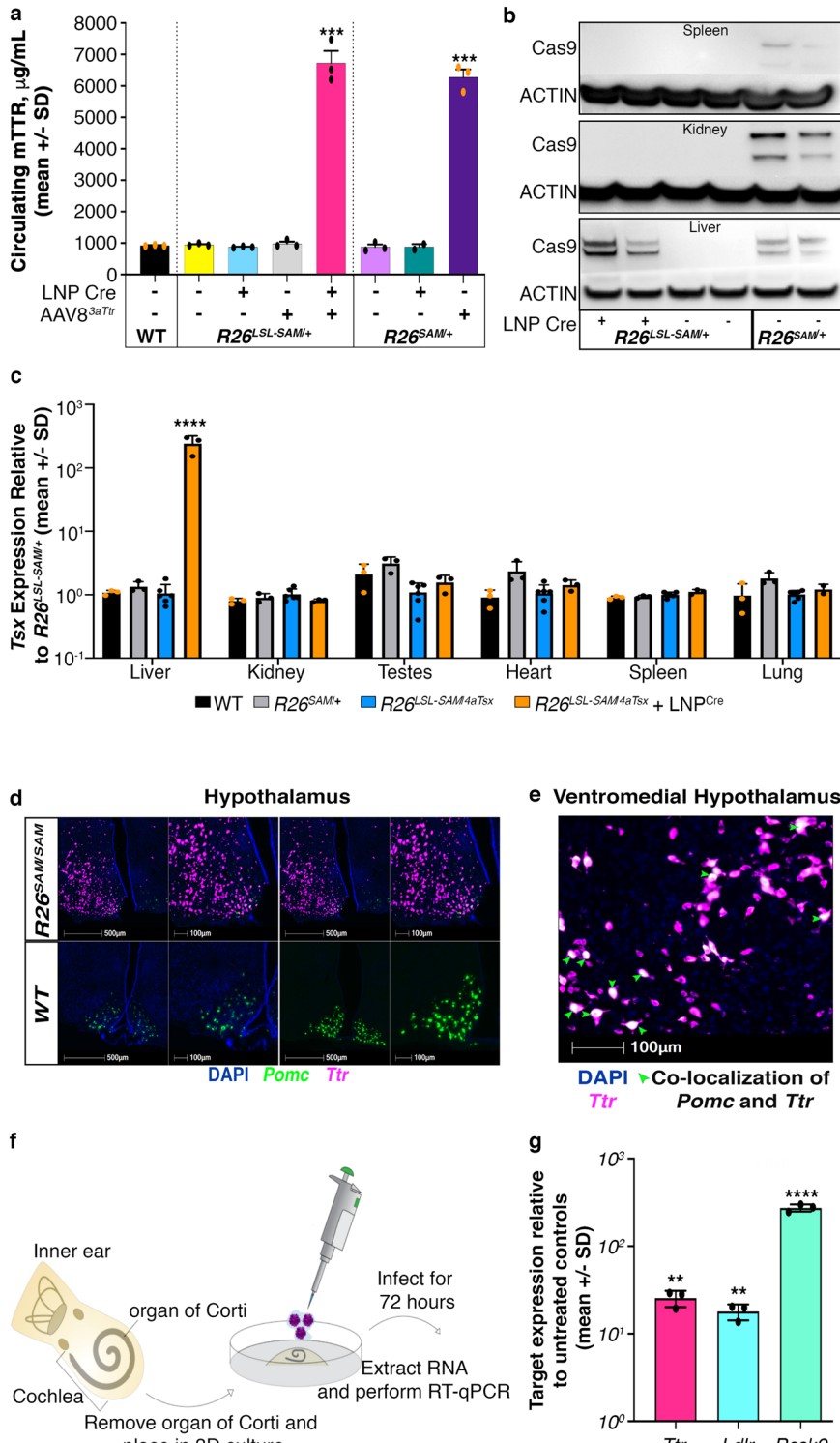

**Fig. 4 Tissue-specific overexpression in vivo. a** LNP[Cre] was delivered to $R26^{LSL-SAM/+}$ to activate SAM expression in the liver ($n = 3$); data was plotted as the mean per group $+/-$ SD. $p$-values of each comparison can be found in Supplementary Table 4. Three replicates were conducted with similar results. **b** A western blot showing dCas9[SAM] expression (219 kDa) in $R26^{LSL-SAM/+}$ tissues treated with LNP[Cre]. dCas9[SAM] signals are normalized to actin (41 kDa). One repeat was completed with similar results. **c** Expression profile of $Tsx$ in $R26^{LSL-SAM/4aTsx}$ tissues with and without LNP[Cre] treatment. $Tsx$ expression relative to $R26^{SAM/+}$ was determined by RT-qPCR ($n = 3$) and plotted as the mean per group $+/-$ SD; $p = 0.000082$. **d** RNAscope of hypothalamus sections showing DAPI (blue), $Pomc$ (green) and $Ttr$ (magenta) in $R26^{SAM/+}$ and WT mice ($n = 2$). **e** Zoom of ventromedial hypothalamus showing co-localization of $Pomc$ and $Ttr$. Green arrows are pointing to white patches that represent co-localization. Additional individual repeats have not been conducted. **f** Graphical protocol for harvest and culture of the organ of Corti. **g** RT-qPCR determination of $Ttr$ ($p = 0.0015$), $Ldlr$ ($p = 0.001395$), and $Pcsk9$ ($p = 0.000052$) expression levels in organ of Corti explant cultures to $R26^{SAM/SAM}$. Results are plotted as the mean per group $+/-$ SD ($n = 3$). Asterisks (*) indicates significance, and the number of asterisks indicates the number of 0s after the decimal point. One-tailed, unpaired Student's $t$-test for (**a**, **c**, **g**).

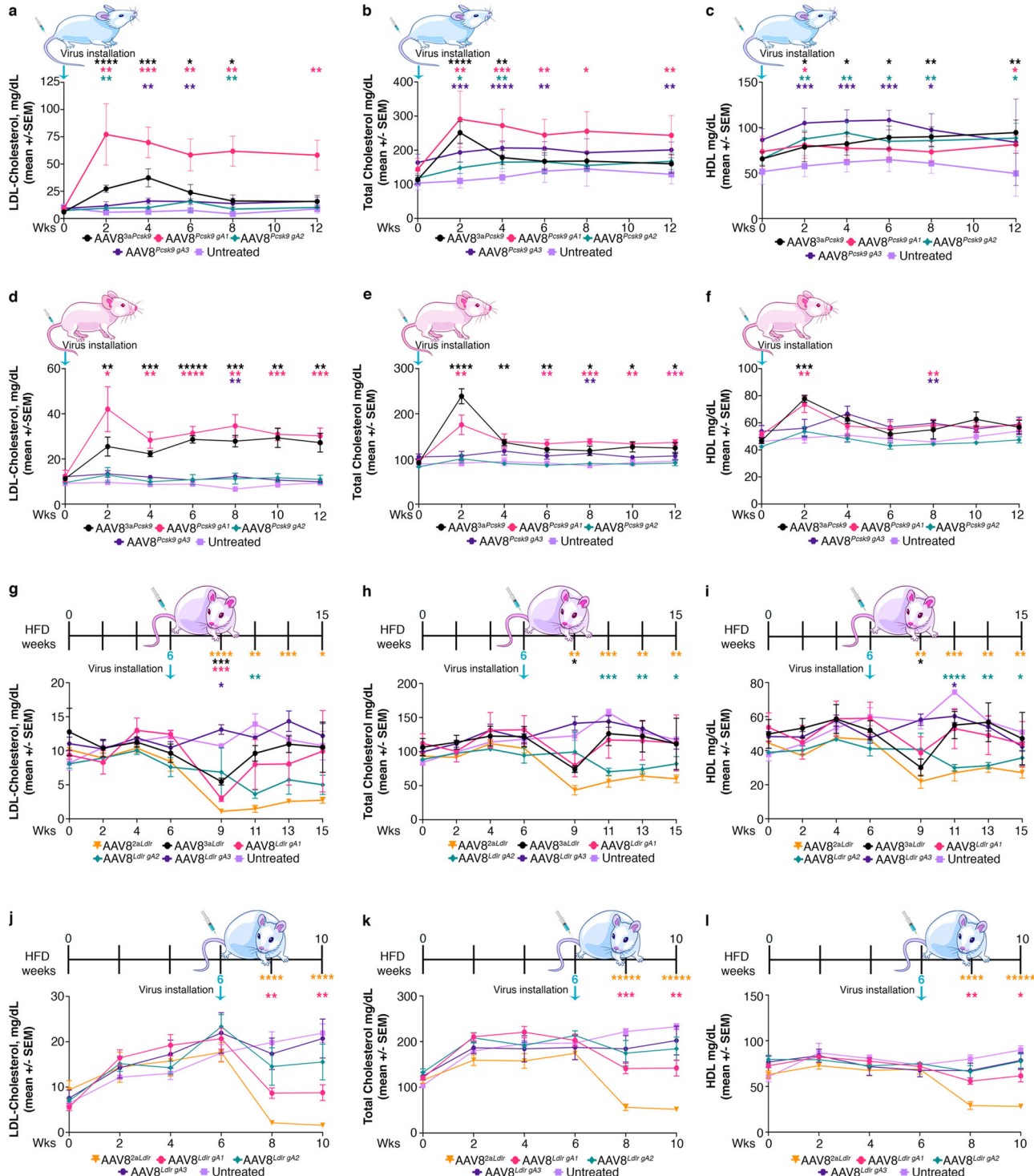

**Fig. 5 Bi-directional modulation of lipid metabolism. a–c** *Pcsk9* activating guides were introduced by AAV8 to male *R26[SAM/SAM]* mice after a baseline bleed ($n = 5$) with bi-weekly bleeds taken thereafter for serum chemistry. Weekly *p*-values can be found in Supplementary Table 5. **a** LDL-cholesterol, **b** total cholesterol, and **c** HDL-cholesterol levels were plotted as mean $+/-$ SEM per group. **d–e** *Pcsk9* activating guides were introduced by AAV8 to female *R26[SAM/SAM]* mice after a baseline bleed ($n = 5$) with bi-weekly bleeds taken thereafter for serum chemistry. Weekly *p*-values can be found in Supplementary Table 5. **d** LDL-cholesterol, **e** total cholesterol, and **f** HDL-cholesterol levels were plotted as mean $+/-$ SEM per group. **g–i** Homozygous female *R26[SAM]* study mice were placed on a HFD for six weeks prior to AAV8[Ldlr] activating viruses were introduced ($n = 4$). Weekly *p*-values can be found in Supplementary Table 5. **g** LDL-cholesterol, **h** total cholesterol, and HDL-cholesterol levels were plotted as mean $+/-$ SEM per group. (j-l) Homozygous male *R26[SAM]* study mice were placed on a HFD for 6 weeks prior to AAV8[Ldlr] activating viruses being introduced ($n = 5$). Weekly *p*-values can be found in Supplementary Table 5. **j** LDL-cholesterol, **k** total cholesterol, and **l** HDL-cholesterol levels were plotted as mean $+/-$ SEM per group. Statistics: Asterisks (*) indicates significance, and the number of asterisks indicates the number of 0s after the decimal point. One-tailed, unpaired Student's *t*-test for (**a-l**).

**Fig. 6 In vivo applications of $R26^{SAM}$ gene modulation. a** Schematic of murine *Ttr* replaced with WT human TTR sequence. Mouse sequences are represented by blue lines and human sequences by purple. **b** Circulating mouse and human TTR levels were determined by species-specific ELISA and plotted as mean +/− SD ($n = 8$). Commercial normal human serum (NHS) was run as a control for normal human expression levels ($n = 1$). Humanized allele vs NHS: $p = 0.0071$. **c** Homozygous humanized TTR mice were crossed to $R26^{SAM/SAM}$ mice and injected with AAV8$^{3aTtr}$ or AAV8$^{GFP}$ ($n = 4$). Asterisks (*) indicates significance, and the number of asterisks indicates the number of 0s after the decimal point. One-tailed, unpaired Student's *t*-test for (**b**, **c**).

greater magnitude of reduction in HDL and total cholesterol than AAV8$^{3aLdlr}$ (Fig. 5h, i). No substantial difference was noted in regard to triglycerides (Supplementary Fig. 7j). To further validate that this SAM mouse model is effective in both genders, the AAV$^{2aLdlr}$ and individual guides were administered to male mice. As observed with *Pcsk9* modulation, $R26^{SAM/SAM}$ alleles are similarly effective in males and females (Fig. 5j–l and Supplementary Fig. 7k).

*Transthyretin* amyloidosis (ATTR) is characterized by the buildup of amyloid deposits in multiple organs and is typically associated with mutations in the *TTR* gene[55,56]. We sought to model this disease with an allelic series beginning with the WT human coding sequence targeted to, and expressed by, the endogenous mouse locus (*Ttr$^{hu/+}$*; Fig. 6a). After breeding to homozygosity, the humanized mouse locus failed to express hTTR protein at levels on par with humans (Fig. 6b). To tailor this model more closely to human physiologic expression, *Ttr$^{hu/hu}$; R26$^{SAM/SAM}$* mice were generated. Four males and four females were injected with AAV8$^{3aTtr}$ and a 27-fold increase in circulating hTTR protein was observed relative to baseline (Fig. 6c). No significant increase in hTTR was observed in animals injected with control AAV8$^{GFP}$. Thus, SAM activation was able to sufficiently increase WT hTTR expression in humanized mice as the first step to allow us to move forward in the generation of an ATTR mouse model.

## Discussion

The development of CRISPRa has enabled reproducible over-expression that can be used to investigate signaling pathways and to generate disease models. We demonstrated that $R26^{SAM}$ mice can induce temporal and spatial gene activation in vivo by exploiting various delivery modalities to male and female mice. The delivery, dosage, and gRNA selection provide several choices to fine-tune the target expression for any gene of interest, including humanized alleles. Targeting guide arrays into the second $R26$ allele facilitates long-term in vivo target activation. We also showed that Cre-mediated, tissue-specific expression of SAM can be leveraged to reduce off-tissue gene activation. Furthermore, SAM technology can simultaneously activate multiple genes in vivo using a single viral vector encoding different gRNAs. Lastly, we demonstrated the ability to generate an in vivo hypercholesteremia disease model with CRISPRa-mediated control of physiologic LDL levels.

The gRNA delivery method can be selected to support the desired timeline and dynamic range of the study. To illustrate this, consider the modulation of *Ttr* expression using a targeted allele, AAV, LNP, or HDD. Each approach has a unique activation range and duration. A targeted expression array will promote

stable expression but may add a year or more to generate the mouse model (Fig. 7a). Random transgenic insertion of gRNA expression arrays has not yet been evaluated but would add months to model generation (Fig. 7b). Non-transgenic methods to introduce gRNAs, such as LNP and AAV, have greatly accelerated our ability to create novel disease models (Fig. 7c). Although widely used to transduce hepatocytes, HDD activation facilitated the mildest increase in expression. These data show that LNP and viral delivery methods are capable of stronger upregulation than conventional HDD and have the added benefit of providing greater control over the anticipated expression level. LNP studies can be extended or enhanced with additional dosing of LNP particles. The successful redose of animals at a single timepoint without adverse effects suggests that sequential dosing of the same animals may be viable. While the mechanism for sustained LNP activation requires further investigation, these data suggest that the upregulation potential of LNP delivery may be more impactful and less transient than initially anticipated.

While viral gRNA delivery methods can be tissue-specific based on the chosen serotype, we further demonstrate that specificity can be mediated by conditional SAM expression. We confirmed that, even with active gRNA expression, only $R26^{LSL-SAM}$ tissues treated with Cre display target activation. The rapidly expanding field of transdifferentiation typically relies on viral delivery of one or more cDNAs[57]. We believe these endeavors can benefit from the use of SAM with an integrated or single viral array that can activate two or more genes in the same cell. This could be a substantial step forward in transdifferentiation studies requiring the simultaneous upregulation of multiple genes.

Gene modulation using the SAM system is a powerful tool for disease modeling as illustrated by our 12–15-week generation of a hypercholesterolemia model in both male and female mice. It may be possible to expand the severity of our phenotype by identifying functional gRNAs to replace AAV8$^{Pcsk9-gA2}$ and AAV8$^{Pcsk9-gA3}$ to potentially increase total cholesterol levels beyond the 300 mg/dL achieved here. This highlights the need to develop a reliable method to screen gRNA efficiency prior to building an array. Tangentially, the implications of *Ldlr* over-expression by SAM highlight the possibility of targeted *Pcsk9* downregulation by CRISPRi as a potential therapeutic. The knockout of genes is fairly common and attainable, but tunable and tissue-specific gene expression reduction (as opposed to complete elimination) could be a powerful alternative.

In addition to generating new disease models, we further validated that SAM can improve upon existing models by increasing hTTR levels in a humanized mouse model by 27-fold with AAV8. We also demonstrated that $R26^{SAM/3aTtr}$ yielded a stable fourfold increase in circulating protein, a level that would

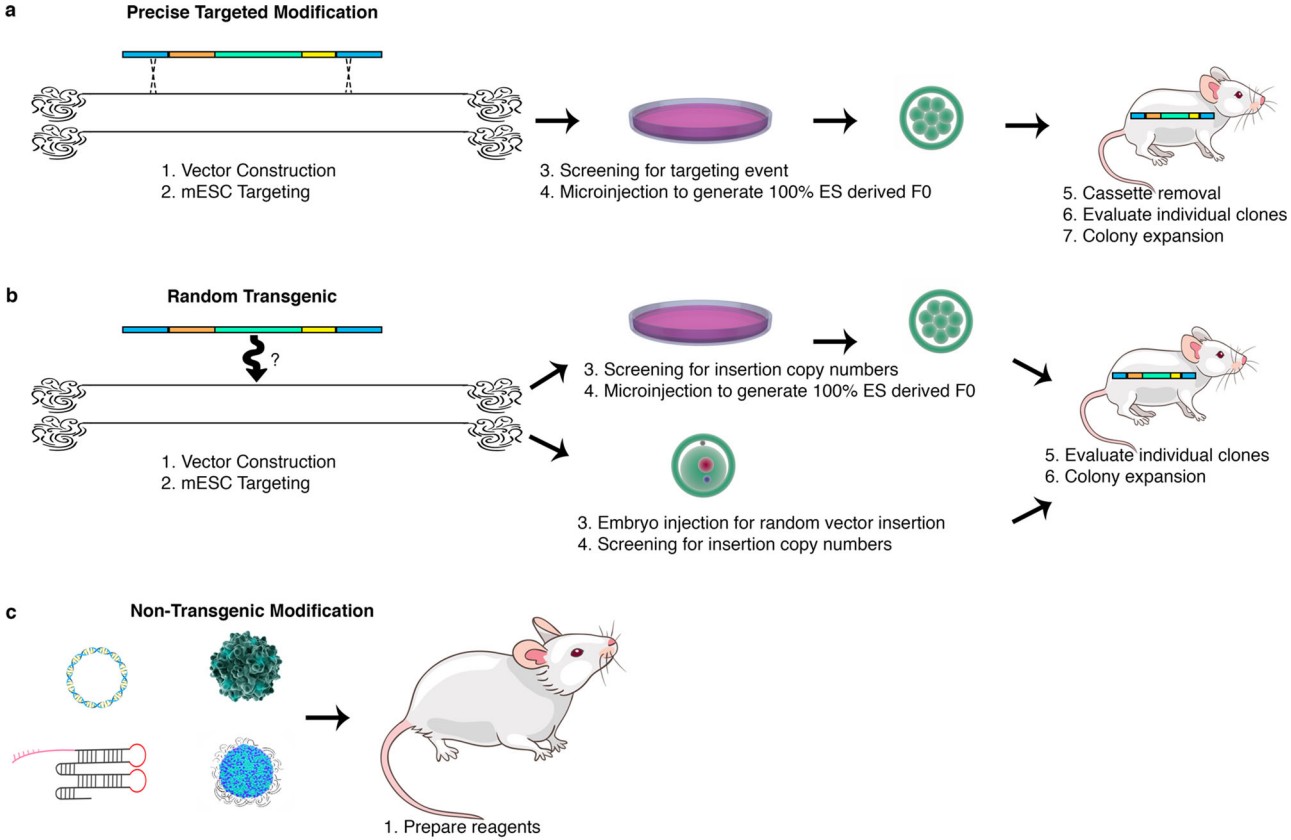

**Fig. 7 Overview of methods used to generate in vivo overexpression models**. Timeline of overexpression allele generation by **a** targeted mESC, **b** random transgenic in mESC or embryos, and **c** tissue-specific, non-transgenic, delivery of guides.

be on par with human expression levels. What's more, using LNP, we were able to facilitate activation levels between six- and ninefold dependent on the amount of gRNA packaged. Therefore, SAM activation has the potential to be tuned to a more representative level of normal human expression or exacerbated above the disease state.

While activation by a SAM allele does not induce excessive perturbations of the genome in mESC or live mice, caution should be exercised when a target gene is in a head-to-head orientation with its neighbor. As illustrated by *Rs1* and *Ppef1*, collateral activation of adjacent genes in a head-to-head context may occur. As this was observed in one of five targets, more data is required to determine the cause and frequency of this off-target effect. The mouse genome contains 1071 head-to-head gene pairs (corresponding to 2130 unique genes) with 400 bp or less between their respective TSS[58]. As such, it will be important to understand which factors are involved in driving collateral activation. Individual *Rs1* activating gRNAs (including those found in the array) could be employed to determine if (1) a common enhancer element was activated (2) gRNA synergy is needed or (3) limiting the gRNAs could ameliorate the phenomenon. Regardless of the delivery approach, the number, position, and quality of gRNAs included in an array can be used to modulate the activation of a target or potentially impact which splice variant is activated. Alternative isoforms have been implicated in cancers, cardiovascular disease, diabetes, and many neurological disorders[59]. The ability to overexpress specific isoforms could be an improvement over traditional overexpression techniques and may lead to novel discoveries. Further validation of the impact on various isoforms and application to specialty techniques, such as transdifferentiation, warrant further studies.

## Methods

**Generation of SAM mESC**. Targeting of mESC (50% C57BL/6NTac and 50% 129S6/SvEvTac) was performed using previously described methods[24]. Briefly, a large targeting vector was built by modifying the *R26* BAC (BAC_ESr2-445b1_sfi_1) to replace 0.081 kb of *R26* intron one with 9.1kb encoding neomycin selection (amino 3'-glycosyl phosphotransferase) and tandem polyadenylation signals flanked by LoxP sites (LoxP-Stop-LoxP or LSL) followed by dCas9 fused to VP64. A P2A peptide also links a fusion of MCP, HSF1, and P65 to dCas9-VP64 such that the single transcript can be driven by the *R26* promoter. The linearized modified BAC was electroporated into mESCs to drive homologous recombination at the *R26* locus utilizing the 11kb and 40kb targeting arms from the modified BAC. Positive transformants were selected with Neomycin resistance. Transgenic insertions were distinguished from targeted recombinations based on quantitative polymerase chain reaction (qPCR)[24]. Initially, expression of the SAM allele is blocked by the LSL cassette. Once targeting was confirmed, the selected clone was electroporated with Cre recombinase to excise the LSL blocking cassette and generate the active allele.

**Mouse production**. SAM mESCs were injected into eight-cell embryos to generate 100% ES-derived F0 mice for in vivo expression validation[24,60]. Briefly, a small hole was created in the zona pellucida of Swiss Webster embryos to facilitate the injection of targeted mESCs. Injected eight-cell embryos were transferred to surrogate mothers to produce live pups carrying the transgene. Upon gestation in a surrogate mother, the injected embryos produce F0 mice that carry no detectable host embryo contribution. The fully mESC-derived mice are generally normal, healthy, and fertile (with germline transmission)[60]. All animal experiments were authorized and performed in accordance with the guidelines for the Institutional Animal Care and Use Committee (IACUC) at Regeneron.

**Design of gRNA**. gRNAs were designed using UCSC with reference to CRISPOR and BLAT[33,37,61]. gRNAs were designed in the 200–300 bases upstream of the TSS unless annotated enhancers were targeted (as in the case of *Tsx*). All sequences can be found in Supplementary Table 14.

**Generation of targeted gRNA arrays**. *R26* integrated arrays were composed of three or more gRNA expression units, each driven by a U6 promoter, with the SAM MS2 loop extension tracr variant (tracr[MS2]). Each unit was followed by an

extended RNA Polymerase III termination sequence. A puromycin (3′-deoxy-N,N-dimethyl-3′-[(O-methyl-L-tyrosyl)amino]adenosine) resistance gene with tandem polyadenylation signals flanked by Rox sites (Rox-Stop-Rox or RSR) was integrated upstream of the array to avoid R26 driven transcripts disrupting gRNA expression. As in the generation of targeted SAM mESC, the arrays were first introduced to the R26 BAC and then targeted into either R26$^{LSL-SAM/+}$ or R26$^{SAM/+}$ mESC.

**Generation of viral particles.** gRNA sequences (single and array) were cloned into the appropriate AAV backbones by standard ligation. HEK 293T cells were maintained in Dulbecco's modified Eagle's medium (DMEM), GlutaMAX supplemented with 10% fetal bovine serum (FBS), penicillin and streptomycin, and nonessential amino acids (Thermo Fisher Scientific). Serum was reduced to 1% FBS for transfection when virus was to be concentrated by polyethylene glycol (PEG).

AAV8 vectors were produced by transient transfection of HEK 293T cells. Transfections were performed using Polyethylenimine (PEI) MAX (Polysciences). Cells were transfected with three plasmids encoding adenovirus helper genes, AAV2 rep and AAV8 cap genes, and recombinant AAV genomes containing transgenes flanked by AAV2 inverted terminal repeats (ITRs). Virus containing medium was collected and filtered through a 0.2 μm PES membrane (Nalgene). Virus was either purified by a series of centrifugation steps or density gradient ultracentrifugation.

For purification by centrifugation, virus containing medium was concentrated by PEG precipitation as previously described[62]. The pellet was resuspended in 1xPBS (Life Technologies) and further clarified by centrifugation at 10,000 RCF. Supernate was transferred and AAV was pelleted in an ultracentrifuge at 149,600 RCF for 3 hours at 10 °C. The AAV containing pellet was resuspended in 1xPBS, clarified by centrifugation, and filtered through a 0.22 μm cellulose acetate membrane (Corning).

For purification by iodixanol gradient separation, medium was concentrated by tangential flow filtration and loaded onto an iodixanol gradient. Iodixanol solutions and gradients were prepared with slight modifications as previously described[63]. Gradients were spun at 149,600 RCF for 14 hours in an ultracentrifuge. The AAV containing fraction was extracted and buffer was exchanged into 1xPBS with 0.001% Pluronic (Thermo Fisher Scientific) using Zeba Spin Desalting columns (Thermo Fisher Scientific).

**Lipid nanoparticle formulation.** Stock solutions of (6Z, 9Z, 28Z, 31Z)-heptatriaconta-6,9,28,31-tetraen-19yl 4-(dimethylamino)butanoate (MC3; Biofine), 1,2-distearoyl-sn-glycero-3-phosphocholine (DSPC; Avanti), cholesterol (Chol; Avanti), and 1,2-Dimyristoyl-sn-glycero methoxypolyethylene glycol (PEG-DMG (2000); NOF) were prepared at 50mM in Ethanol. These lipids were mixed to yield a molar ratio of 50:10:38.5:1.5 (MC3:DSPC:Chol:PEG-DMG). The gRNA or mRNA was prepared in 10mM sodium citrate (pH 5) to 225 μg/mL. Through the use of microfluidic mixing of the BenchTop Nanoassemblr (Precision Nanosystems), the gRNA or mRNA and lipids were mixed at 12 mL/min flow rate and at a 3:1 volumetric ratio of (gRNA or mRNA):Lipids. LNPs were diluted into PBS (pH 7.4) to dilute the ethanol and subsequently concentrated using a centrifugal filter (Amicon, 10 kDa cutoff). The gRNA or mRNA was quantified through a modified Ribrogreen assay (Life Technologies), briefly the LNPs were quantified in TE and TE with 2% Triton X-100 and then the protocol was performed as written. The total encapsulated gRNA or mRNA was determined by the measurement of mRNA in the Triton-X-100 sample (Total mRNA)—TE sample (free mRNA). Prior to delivery to animals the LNPs were filtered through a 0.22 μm syringe filter and diluted to the appropriate concentration in PBS (pH 7.4) at a total volume for i.v. injection of 200 μL.

**Tail vein injection.** The lateral tail vein was injected by inserting a 27-gauge needle into the vein at the base of the tail and injecting ~1–2 × 10$^{11}$ viral genomes in 100 μL.

**Lipid profile.** All lipid profiles were completed on the ADVIA Chemistry XPT system (tailored to mouse levels). Study animals involved in Ldlr expression modulation were placed on Research Diets Inc., Rodent Diet 60 kcal HFD (D23492) formulated with 245g of Lard and 25g of soybean. The PCSK9 ELISA was performed utilizing the Mouse PCSK9 ELISA Kit (ab215538), using serum from multiple timepoints of the Pcsk9 guide treated and untreated animals at 2 dilution ranges 1:1000 and 1:5000 performed as recommended by the manufacturer. N = at least 3 mice for each timepoint.

**Hydrodynamic delivery.** Fifty micrograms of plasmid coding for mouse Ttr, 3aTtr, and empty vector were diluted in sterile saline (0.9% NaCl) to a volume that is ~10% of the mouse body weight and injected into the tail vein of 9–12-week-old male SAM mice over 6 to 10 seconds[64].

**Stereotactic surgery.** Surgical procedures followed aseptic rodent surgery guidelines. Mice were anesthetized with isofluorane gas and mounted in a stereotaxic apparatus (Kopf Model 963). An incision was made to the scalp exposing bregma. Following a craniotomy, a microinjector (WPI Inc. UMP-3T) delivered 500nl of

either control virus (AAV8-GFP) or SAM (AAV8-TTR-SAM) into arcuate (AP: −1.5, ML: −0.2, DV: 5.75) and parietal cortex (AP: −1.5, ML: .02, DV: 1.0) respectively at a rate of 200nl/min. Coordinates were chosen using the Paxinous brain atlas as reference. The needle (35G-beveled, Nanofil) was allowed to settle for 2 min prior to injection and removed 8 min post injection. Bone wax was used to fill the craniotomy and the surgical incision was closed with wound clips.

Five weeks later, mice were anesthetized (110 mg/kg Nembutal, i.p.) and transcardially perfused with 4% paraformaldehyde in 0.1 M sodium borate buffer at 4 °C. Brains were post-fixed for 2 h at 4 °C and then incubated in 0.05 M potassium-PBS (K-PBS) containing 15% (w/v) sucrose 4 °C for 12–16 h. Brains were sectioned on a sliding microtome (30 μM), collected in equally-spaced series and stored in cryoprotectant (20% glycerol and 30% ethylene glycol in 0.1 M phosphate buffer) at −20 °C[65].

**RNAscope.** Freely floating brain series were pretreated with 1% H$_2$O$_2$ in K-PBS and 0.3% glycine and 0.3% (v/v) Triton X-100 in K-PBS for 10 min. They were then mounted on Superfrost Plus Gold Slides (EMS, #71864-01). Fluorescent in situ hybridization was conducted using ACD RNAScope Fluorescent Multiplex Kit V2 following the manufacturer protocol. Ttr and Pomc were labeled with Opal 690 Reagent Pack (Perkin-Elmer) and Opal 520 Reagent Pack (Perkin-Elmer) respectively. The slides were counterstained with DAPI and mounted with Prolong Gold Mounting Media (Invitrogen), then cover slipped. Slides were scanned with a 20x objective Hamamazu camera (Axio Scan.Z1) and images were post-processed in Halo.

**TTR ELISA.** Mouse TTR secretion into serum was quantified via ELISA, according to the manufacturer's protocols (Aviva Systems Biology: Mouse Prealbumin ELISA Kits (OKIA00111). The samples were diluted 1:1,000, 1:10,000, and 1:30,000 in diluent buffer and compared to protein standards. Optical density was measured at 450 nm. A standard curve was generated by log-transforming the data and performing a linear regression of the TTR concentration versus the optical density.

**LacZ staining.** The desired tissues were incubated in a 0.2% glutaraldehyde, 4% paraformaldehyde for 30 min at 4 degrees with agitation. After fixation, tissues were washed 3 times in PBS for 20 min at 4 degrees with agitation. Next, tissues were incubated in X-gal (1 mg/mL) staining solution for 2 days at 4 degrees with agitation. Tissues were then washed in PBS and post-fixed in 4% paraformaldehyde overnight with agitation. Paraformaldehyde was washed off with PBS prior to imaging.

**Western blotting.** Cells were grown to a high confluency (>70%) on gelatin coated TC-treated 6-well plates and lysed in RIPA Lysis and Extraction Buffer (Thermo Scientific) with complete protease inhibitors (Thermo Fisher). Lysates were centrifuged at 13,200 RPM at 4 °C for 15 min and transferred to a 1.5 mL Protein LoBind Tube (Eppendorf). Protein was normalized using BCA Protein Assay Kit (Pierce) and loaded onto a 4–12% polyacrylamide gradient gel (Invitrogen) and run in SDS running buffer. Gels were electroblotted onto PVDF membranes (Invitrogen iBlot 2 Transfer Stacks). Membranes were blocked overnight at 4 °C in 5% milk powder in TBS-T (Pierce), before being incubated for at least 4 hours with primary antibody against S. pyogenes Cas9 (1:1000: (Invitrogen Cat#MA5-23519). HRP-conjugated anti-beta-actin (1:1000: Millipore Sigma Cat#MAB1501) antibodies were used as loading controls. After washing in TBS-T, membranes were incubated with HRP-conjugated secondary antibodies for 1 hour. Signals were detected using Novex ECL Chemiluminescent Substrate Reagent Kit (Invitrogen) and images were captured on an Azure c600. Captured images were processed in Adobe Photoshop.

**RT-qPCR of mESC.** mRNA was extracted from cells using Zymo Research Direct-zol RNA miniprep plus kit. DNA was eliminated using the DNase treatment as outlined in the Zymo protocol. RNA from each sample was mixed with the appropriate probes and master mix from the Qiagen QuantiNova Probe RT-qPCR kit. Gene expression was normalized to the expression of the internal housekeeping gene B2m. Oligos can be found in Supplementary Tables 15 and 16.

**RT-qPCR of mouse tissues.** mRNA (up to 2.5 μg) was reverse-transcribed into cDNA using SuperScript® VILO™ Master Mix (Invitrogen). cDNA was amplified with the SensiFAST Probe Hi-ROX (Meridian) using the ABI 7900HT Sequence Detection System (Applied Biosystems). B2m was used to normalize any cDNA input differences. Data was reported as the comparative CT method using delta delta CT.

**RNA-seq RNA preparation.** Total RNA was purified from all samples using MagMAX™-96 for Microarrays Total RNA Isolation Kit (Ambion) according to manufacturer's specifications. Genomic DNA was removed using Qiagen DNase kit. mRNA was purified from total RNA using Dynabeads mRNA Purification Kit (Invitrogen). Strand-specific RNA-seq libraries were prepared using KAPA mRNA-Seq Library Preparation Kit (Kapa Biosystems). Twelve-cycle PCR was

performed to amplify libraries. Sequencing was performed on Illumina HiSeq®2500 (Illumina) by multiplexed single-read run with 33 cycles.

**RNA-seq read mapping and statistical analysis of differentially expressed RNA**. Raw sequence data (BCL files) were converted to FASTQ format via Illumina bcl2fastq v2.17. Reads were decoded based on their barcodes and read quality was evaluated with FastQC (http://www.bioinformatics.babraham.ac.uk/projects/fastqc/). Reads were mapped to the mouse genome (NCBI GRCm38) using ArrayStudio® software (OmicSoft®, Cary, NC) allowing two mismatches. Reads mapped to the exons of a gene were summed at the gene level. Genes were flagged as detectable with minimum 10 reads. Differentially expressed genes were identified by DESeq2 package and significantly perturbed genes were defined with fold changes no <1.5 in either up or down direction and with $p$-values of at least 0.01. Genes with 0.5 TPM or less in all replicates of the activator and the control were excluded before log transformation. Finally, Zero Transformation was performed on all remaining zero values for the purpose of plotting on $Log_2$ scaled correlation graphs.

**Explant culture**. Neonatal organs of Corti (P0-P5) were harvested from $R26^{SAM/SAM}$ and microdissected for culture in DMEM/F-12 +7% FBS for 24 hours. Culture media was replaced with fresh culture medium containing 1 × $10^{11}$ viral genomes in a final volume of 200 μL. Virus containing media was left on the cells for 72 hours and then tissues were collected for RNA extraction and RT-qPCR analysis.

**Statistical analysis**. All of the data are presented as the mean plus or minus standard deviation (+/− SD) or standard error of mean (+/− SEM). For in vivo experiments, $n$ = number of animals. Statistical analyses were carried out using GraphPad Prism 8 Software. All details of statistical tests, including $p$-values, can be found in the Supplementary Tables.

**Reporting summary**. Further information on research design is available in the Nature Research Reporting Summary linked to this article.

## Data availability

Source data associated with this study are present in the paper or the Supplementary Materials. Raw RNAseq data can be accessed from BioProject ID PRJNA669145 (https://www.ncbi.nlm.nih.gov/bioproject/PRJNA669145). Source data are provided with this paper.

## Materials availability

Materials described in this manuscript are available by contacting Regeneron Pharmaceuticals, Inc. (email address: preclinical.collaborations@regeneron.com) for academic and non-profit purposes only under an MTA, which allows the use of mice for academic but not commercial purposes.

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

## Acknowledgements

We acknowledge the contribution of the entire VelociGene team for generation of constructs and mice. We acknowledge the ENCODE Consortium, ENCODE/LICR (wgEncodeEM002593, wgEncodeEM002490, wgEncodeEM002594), ENCODE/Caltech (wgEncodeEM002130, wgEncodeEM002132, wgEncodeEM002117, wgEncodeEM002120) and ENCODE/University of Washington (wgEncodeEM001932) for generating the Tsx datasets. We thank members of the Regeneron Core Services for support and material generation. We thank Hock E for HDD support. We thank Rostislav Chernomorsky, Jennifer Espert, Peng Tham, Qian Tang, and Andrew Romano for RNA extraction and qPCR processing of mouse tissue samples. We thank Anthony Reed Jr. for graphic support. We thank Susannah Brydges, Jeremy Rabinowitz, and Aris Economides for critical reading of the manuscript.

## Author contributions

Conceptualization and supervision of the project was done by C. Hunt, S.H., E.C., B.Z., and G.G. Experimental design was carried out by C. Hunt, S.H., E.C., B.Z., and G.G. Experiments were carried out by C. Hunt, S.H., D.W., T.H., C. Herman, J.W., H.B., R.M., H.N., K.C., S.C., S.M.T., J.B., P.M., and G.D. Computational analyses of RNA-seq data was carried out by Q.S. and Y.X. Data analysis was carried out by C. Hunt, S.H., D.W., E.P., J.A., J.H., M.D., D.F., E.C., B.Z., and G.G. The original draft was written by C. Hunt, with later edits and reviews by S.H., E.P., E.C., D.F., B.Z., and G.G.

## Competing interests

C. Hunt, S.H., D.W., E.P., T.H., C. Herman, J. W., H.B., Q.S., D.V., J.A., K.C., J.H., S.C., M.D., S.M.-T., M.P.M.-P., G.D., D.F., E.C., B.Z., and G.G. are employees of Regeneron Pharmaceuticals Inc ("Regeneron"). Regeneron has filed patent applications around the described work. The remaining authors declare no competing interests.
