## [Peer Review File · Nature Communications]

Reviewers' Comments:

Reviewer #1:

Remarks to the Author:

This is a well-done study describing a new CRISPRa mouse model with extensive characterization and validation with different gene targets, delivery methods, and target tissues. It will certainly be of value to the broad research community. The demonstration of gene activation by LNP delivery is particularly innovative and compelling. There are some issues that should be addressed before publication, with the major issue being that this type of resource article, which doesn't provide any new biological insights, is only useful to the community if the mouse models and constructs are shared broadly, but there are no details about availability or resource sharing provided here. That should be addressed and confirmed prior to publication.

Major issues:

1. Have the mice and plasmids been deposited at a repository for non-profit research? There are no new biological insights in this study. The resources described could be very valuable, but only if they are shared.
2. In general, statistical tests are missing throughout the paper.
3. The disease modeling and activation of TTR is very interesting, but the authors don't seem to demonstrate that the mice with elevated hTTR actually get the buildup of abnormal amyloid deposits as is the goal for disease modeling, as described in the text. Can they verify that the disease model leads to the disease phenotype?
4. Gene activation with CRISPRa by LNP delivery of RNA is very exciting for potential therapeutic applications, but really only if redosing is possible. Can the authors show that gene targets can be reactivated after transient activation? Though outside the scope of the study, if the authors should show similar results in WT mice with dCas9-SAM mRNA delivery, that would be really exciting.
5. The study is mostly based on delivering arrays of gRNAs, but it is well known that having these repetitive sequences in tandem (both promoters and tracrRNA constant regions) can lead to recombination – especially of the AAV vectors. How were arrays validated? Can the authors demonstrate lack of recombination in this format?

Minor issues:

1. The introduction would benefit from discussion of the many other dCas-based activators that have been described and why they chose the SAM system.
2. First paragraph of results – there is a lack of experimental details. What cells? how were clones characterized? How was Cre delivered?
3. Regarding the effect on neighboring genes (fig 1) – is there any evidence of looping between promoters from public data sets (promoters acting as enhancers)?
4. Fig S1a – is targeting the Tsx enhancer critical? It would be helpful to show genomic tracks/evidence of enhancer activity.
5. Fig S1e-g – is this effect on neighboring genes statistically significant?
6. Pg 4 0 is this a typo? – “R26SAM/3aTtr mice maintained circulating TTR at a level that was 2.5-fold higher than in R26SAM/3aTtr + or WT mice (Fig. 2a).”

7. AAV8 is referred to as "liver specific" but this is not the case. It will also go to muscle and heart, and the authors should assess these tissues as well.

8. Fig 3a – why is there loss of TTR activation over time? AAV8 in the mouse liver is generally thought to be episomally stable. The authors could assess this by qPCR for vector genomes.

9. In Figure 3 – the panels change between homozygous and heterozygous mice- what are the differences between these models and why did the authors change between experiments?

10. Figures 5, S6, and S7 are all missing important statistical tests. In particular, there are claims as to whether HDL, total cholesterol, and triglycerides are or are not changing without statistical support.

11. With respect to differences in AAV vectors with one gRNA or arrays of gRNAs – the authors should confirm equivalent levels of gRNA expression in vivo by qRT-PCR. They should also discuss how AAV titering may have influenced these results.

Reviewer #2:

Remarks to the Author:

This manuscript reports a tissue-specific activation approach using synergistic activation mediator (SAM) CRISPRa system in mice. The reported SAM CRISPRa system is a promising gene activation platform in mouse models. Please see my comments below:

1. The authors did not provide statistical analysis for the data reported in this manuscript. The methods for Statistical analysis only stated that analyses were carried out using GraphPad Prism 8 software. Considering most sample sizes were small (N = 2 – 5), some data would be critical to include appropriate statistical analysis.
2. In a few place, the authors mentioned male mice were used, but mots data did not identify sex. It is important to know whether this approach is sex-specific, or it can be applied to both sexes. If it was sex-specific, this would be one limitation that should be clearly stated. It is not rare to see sex difference using similar approaches. For example, administration of AAV8 encoding PCSK9 gain-of-function mutation showed different response between male and female mice (PMID: 30253291), and AAV-CRISPR method was also reported with sex difference (PMID: 30026278).
3. Figure 2f shows gross images of multiple organs with X-gal staining. Since each organ has many different cell types and has heterogeneous structures, it would be important to also show cross-sections of some organs. Suggest to also include kidney in Figure 2f.
4. Figure 3a: Is this mTTR ELISA in plasma or a specific tissue?
5. There has no necessary information how total cholesterol (Figure S7c), LDL, and HDL cholesterol were measured. What was the content of the high fat diet?
6. Figure 5: LDL cholesterol change was modest. This data would be important to have appropriate statistical analysis. It is unclear why mice shown in Figure 5d did not provide HFD, whereas mice shown in Figure 5e and Figure S7c were fed high fat diet for 6 weeks prior to virus installation, but did not continue the high fat diet feeding after virus installation. Increase of PCSK9 or deletion of LDL receptor in mice would not change plasma total cholesterol profoundly unless a saturated-fat enriched diet was fed.
7. It is unclear why Figure 5 only shows LDL-cholesterol, whereas Figure S7 shows total cholesterol, HDL-cholesterol, and triglycerides.

June 25, 2020

Dear reviewers,

Thank you for your constructive comments to improve the manuscript. We provide below specific responses to the questions and concerns you raised.

Reviewers' comments:

Reviewer #1 (Remarks to the Author):

Major issues:

1. Have the mice and plasmids been deposited at a repository for non-profit research? There are no new biological insights in this study. The resources described could be very valuable, but only if they are shared.

We agree these valuable resources should be made available to the non-profit research community. Due to the double-blind peer review approach, these details did not appear in the manuscript submitted for review. We apologize for any confusion this may have caused. Partial instruction for how materials may be requested have been added in "Data and materials availability" on page 16 and the full instructions will be included in the final publication.

2. In general, statistical tests are missing throughout the paper.

We agree with the reviewer and apologize for this omission. We have now added the appropriate statistical data for all experiments. Please refer to the supplemental data section, "Consolidated metadata for all figures."

3. The disease modeling and activation of TTR is very interesting, but the authors don't seem to demonstrate that the mice with elevated hTTR actually get the buildup of abnormal amyloid deposits as is the goal for disease modeling, as described in the text. Can they verify that the disease model leads to the disease phenotype?

Thank you for bringing this detail to our attention. We did not intend to imply that the WT hTTR allele is the disease model, but rather the first step in creating the disease model whereby we will incorporate disease associated variants into our humanized allele. We agree with the reviewer that this may not have been clear and we have made updates to the section entitled "SAM can modulate existing disease models" in order to clarify that this is a precursor to the disease model. The disease model with incorporated human mutations is in development and will require additional evaluation before it can be considered for publication.

4. Gene activation with CRISPRa by LNP delivery of RNA is very exciting for potential therapeutic applications, but really only if redosing is possible. Can the authors show that gene targets can be reactivated after transient activation? Though outside the scope of the study, if the authors should show similar results in WT mice with dCas9-SAM mRNA delivery, that would be really exciting.

We thank the reviewer for the encouraging feedback. Following the reviewer's suggestion we completed a LNP redosing evaluation and now report that redosing of LNP-gRNA can

successfully reactive a target gene after transient activation (Figure 3f,S5). We agree with the reviewer that an LNP formulation with dCas9-SAM mRNA and guide would be exciting and are actively working towards this goal. However, we will not be including the data in this manuscript as it is outside the scope for the characterization of this animal model.

5. The study is mostly based on delivering arrays of gRNAs, but it is well known that having these repetitive sequences in tandem (both promoters and tracrRNA constant regions) can lead to recombination – especially of the AAV vectors. How were arrays validated? Can the authors demonstrate lack of recombination in this format?

We agree that it is important to demonstrate the integrity of the vectors utilized in this study. We validate all constructs with restriction digest characterization (with cuts in the ITRs) as well as sanger sequencing analysis. The vectors have been fully sequenced and the result summaries are in Supplemental Materials..

Minor issues:

1. The introduction would benefit from discussion of the many other dCas-based activators that have been described and why they chose the SAM system

While we agree that the various other activation platforms are important to discuss, we have opted to limit this topic due to text limitations. We reference Chavez, A. et al. Comparison of Cas9 activators in multiple species. *Nat Methods* 13, 563-567, doi:10.1038/nmeth.3871 (2016) as the impetus for modeling the dCas9-SAM platform (introduction).

2. First paragraph of results – there is a lack of experimental details. What cells? how were clones characterized? How was Cre delivered?

These details have been previously published (reference 24) and we summarized the details in the Materials and Methods.

3. Regarding the effect on neighboring genes (fig 1) – is there any evidence of looping between promoters from public data sets (promoters acting as enhancers)?

We thank the reviewer for this suggestion; however, we do not see such evidence in public data sets. We plan to further investigate this result and will consider a follow-up publication if warranted.

4. Fig S1a – is targeting the *Tsx* enhancer critical? It would be helpful to show genomic tracks/evidence of enhancer activity

While targeting the *Tsx* enhancer is not critical, we demonstrate that this approach was successfully able to upregulate the target gene even though guides were located further outside the transcriptional start than typical. We have referenced the appropriate ENCODE datasets for this enhancer in the acknowledgements; however, it was removed to support the double-blind review. In this new submission, we will keep a portion of the acknowledgements in order to allow reviewers the opportunity to review the purported enhancer region. Further, we have now included a more detailed graphic of the enhancer region in supplemental figure 1.

5. Fig S1e-g – is this effect on neighboring genes statistically significant

The effect on neighboring genes is only significant for *Ppof1*. Significance calculations are now available in the figures (Figures 1, 2, S1 and S3) as well as the supplemental data section, “Consolidated metadata for all figures.”

6. Pg 4 0 is this a typo? – “R26SAM/3aTtr mice maintained circulating TTR at a level that was 2.5-fold higher than in R26SAM/3aTtr + or WT mice (Fig. 2a).”

Yes, this was a typo. We thank the reviewer for alerting us to this error.

7. AAV8 is referred to as “liver specific” but this is not the case. It will also go to muscle and heart, and the authors should assess these tissues as well

We agree with the reviewer that AAV8 is not liver-specific. Our intent was to convey that the liver is a major target of AAV8 but acknowledge there is potential for other tissues to be effected. This has been clarified in the text and potential off-tissue activations are called out in appropriate sections.

8. Fig 3a – why is there loss of TTR activation over time? AAV8 in the mouse liver is generally thought to be episomally stable. The authors could assess this by qPCR for vector genomes

We believe that the slow decline in TTR protein levels may be attributed to hepatocyte turnover. This has now been added to the text and supported by references 42-44.

9. In Figure 3 – the panels change between homozygous and heterozygous mice- what are the differences between these models and why did the authors change between experiments?

We agree this should have been addressed and thank the reviewer for bringing this omission to our attention. We have now added Figures S5a to demonstrate that there is minimal difference in mRNA detected in tissues from *R26^{SAM/+}* and *R26^{SAM/SAM}*. We further verified the alleles functionally using serum TTR as a readout and found that the difference is not statistically significant as shown in Figure S5b.

10. Figures 5, S6, and S7 are all missing important statistical tests. In particular, there are claims as to whether HDL, total cholesterol, and triglycerides are or are not changing without statistical support

We are sorry for the omission of these important details. Appropriate statistics have been incorporated to figures and we report statistically significant changes in LDL, HDL, and total cholesterol. Graphs for these lipids have been moved from supplemental data to Figure 5 with additional replicates and triglyceride data now appearing in Figures S6 and S7.

11. With respect to differences in AAV vectors with one gRNA or arrays of gRNAs – the authors should confirm equivalent levels of gRNA expression in vivo by qRT-PCR. They should also discuss how AAV titering may have influenced these results

We agree that AAV titering can have the potential to affect the results and have now added a supplemental table detailing the titers introduced to study animals. In addition, we completed

qPCR on the harvested tissues using probes specific to the ITR and to the SAM tracer, please find the data in Figure S4d.

Reviewer #2 (Remarks to the Author):

1. The authors did not provide statistical analysis for the data reported in this manuscript. The methods for Statistical analysis only stated that analyses were carried out using GraphPad Prism 8 software. Considering most sample sizes were small ($N = 2 - 5$), some data would be critical to include appropriate statistical analysis

We agree with the reviewer and apologize for this omission. We have now added the appropriate statistical data for all experiments (please refer to the supplemental data section, "Consolidated metadata for all figures") and conducted repeat experiments for *Pcsk9* and *Ldlr* expression modulation. We hope these replicate studies along with statistics can add confidence for the reproducibility of these results.

2. In a few places, the authors mentioned male mice were used, but most data did not identify sex. It is important to know whether this approach is sex-specific, or it can be applied to both sexes. If it was sex-specific, this would be one limitation that should be clearly stated. It is not rare to see sex difference using similar approaches. For example, administration of AAV8 encoding PCSK9 gain-of-function mutation showed different response between male and female mice (PMID: 30253291), and AAV-CRISPR method was also reported with sex difference (PMID: 30026278).

In the first submission, the sex of the study animals was noted for disease models only. We have now entered the sex of mice in each study and confirm that the CRISPRa approach is valid in both male and female *in vivo* studies (Figure S4c). In addition, we have initiated follow up studies for *Pcsk9* and *Ldlr* expression modulation in the opposite gender as appeared in the first version. While we cannot have the full timeline completed by the re-submission deadline, we now show significant lipid modulation by activating *Pcsk9* expression in female mice over a 6-week time course (previously 12 weeks in male mice). We also had initiated a repeat *Ldlr* study (excluding the poor performing 3-guide array) and document significant changes to serum lipids in male mice over 10 weeks (previously 12 weeks in female). These new data can be found in Figure 5, S6, and S7.

3. Figure 2f shows gross images of multiple organs with X-gal staining. Since each organ has many different cell types and has heterogeneous structures, it would be important to also show cross-sections of some organs. Suggest to also include kidney in Figure 2f.

Thank you for this suggestion; we have now included pictures of kidneys in Figure 2f. Unfortunately, the tissues were only whole mount stained and we will not be able to complete sectioning at this time.

4. Figure 3a: Is this mTTR ELISA in plasma or a specific tissue?

Figure 3a is a long-term study where serum was harvested monthly for characterization of circulating serum TTR by ELISA.

5. There is no necessary information how total cholesterol (Figure S7c), LDL, and HDL cholesterol were measured. What was the content of the high fat diet?

The materials and methods section has been updated to indicate that all lipid profiles were completed on the ADVIA Chemistry XPT system (tailored to mouse levels). A 60 kcal HFD with 245g of Lard and 25g of soybean oil was provided to study mice.

6. Figure 5: LDL cholesterol change was modest. This data would be important to have appropriate statistical analysis. It is unclear why mice shown in Figure 5d did not provide HFD, whereas mice shown in Figure 5e and Figure S7c were fed high fat diet for 6 weeks prior to virus installation but did not continue the high fat diet feeding after virus installation. Increase of PCSK9 or deletion of LDL receptor in mice would not change plasma total cholesterol profoundly unless a saturated-fat enriched diet was fed.

Statistical details have been incorporated and we can confirm that the original studies had a significant change in LDL-cholesterol levels. In addition, we conducted repeat studies for both *Ldlr* and *Pcsk9* expression modulation. In the case of *Ldlr*, one study was initiated prior to receiving feedback as we were interested to see if we could have a more impressive change in LDL by eliminating the worst performing guide; these data can be found in Figure 5g-i, S7j. After receiving reviewer feedback, we also conducted studies for both targets in the opposite sex as originally presented. All these repeat studies showed significant LDL cholesterol changes (Figure 5d-f, 5j-l).

All studies involving *Ldlr* activation began with 6-weeks of HFD prior to tail vein injection of AAV particles in order to raise the naturally low LDL-cholesterol level of mice. The HFD was maintained for the full 12-week duration of the *Ldlr* studies and the LDL change was significant at the timepoint following the injection. *Pcsk9* studies did not utilize a HFD. We apologize if the timeline bars associated with the studies were misleading; we have now removed the injection timelines from figures that did not involve HFD and report that the original study was significant at the first timepoint after injection.

7. It is unclear why Figure 5 only shows LDL-cholesterol, whereas Figure S7 shows total cholesterol, HDL-cholesterol, and triglycerides.

We appreciate that it is helpful to see all data in one place. We have now moved the LDL, HDL, and total cholesterol graphs to Figure 5 for both *Pcsk9* and *Ldlr* modulation. Triglyceride results have been placed in supplemental as they are mostly not significant.

Sincerely,

The author

Reviewers' Comments:

Reviewer #1:

Remarks to the Author:

As noted in my original review, this paper is exceptionally technically executed and presents what could be a very valuable resource to the research community. The only major concern is that the mice should be made readily available to the community via a common resource such as Jax. Otherwise, the paper does not report any novel findings and the value and impact would be significantly reduced.

The only other major issue is that in Figures 5, S6, and S7, it seems that they are directly activating PCSK9 and LDLR to increase LDL levels, for which they show the time course of serum LDL levels over many weeks, but they don't show serum PCSK9 levels (the actual target). Why is this? Do they have that data (from the same samples) that they can include? It seems important to understanding the relationship of the target gene with downstream effects.

Other points to address.

1. In the original review, I suggested the authors should assess the levels of gRNA expression by qRT-PCR (point #11). In response, they provided qPCR for DNA vector genomes, which is not the same thing (Fig S4D). My point was that the cassettes in the arrays could likely be interfering with transcription and changing expression levels. Ideally, this would be assessed to understand the value of the array format. At a minimum, the title for Fig S4D should be changed from "Detection of AAV gRNAs..."

2. "The MCP transcription factors interact with the tracr aptamers to synergistically activate targets." – what "synergy" is being referred to here?

3. "Thus, the application of SAM activation to an initially unsatisfactory model has enabled the continued study of ATTR pathogenesis in mouse models." – my understanding is that there is no "pathogenesis" in this model of WT ATTR expression?

4. Fig 6d-f – It's not clear this is necessary. Maybe it should be its own figure since it doesn't relate to Fig 6a-c.

5. Fig S2b – It's confusing what is being measured here. Are they plotting levels of B2m gene expression on a plot that is labelled "dCas9 SAM mRNA"?

Reviewer #2:

Remarks to the Author:

The revised manuscript has been significantly improved.

A few comments:

1. One remaining concern is statistical analysis: The authors state in the figures "T test" and in the tables "Multiple T Test". What kind of t-test was used? For example, Student's t-test or Welch's t-test? Paired or unpaired? One or two tailed? These need to be clarified. "Multiple T test" is a confusing term. Do the authors mean multiple-group comparisons? If so, appropriate multiple-group comparisons such as one or two way ANOVA should be used. Suggest to also provide appropriate justification for the statistical analysis used.

2. Suggest to provide molecular mass for Western blots in Figures 1b, 4b, and S2a.

3. Figure 2f: It appears the images have different magnification. Suggest to include scale bars.

October 16, 2020

Dear reviewers,

Thank you for your constructive comments to improve the manuscript. We provide below specific responses to the questions and concerns raised by you.

REVIEWER COMMENTS

Reviewer #1 (Remarks to the Author):

As noted in my original review, this paper is exceptionally technically executed and presents what could be a very valuable resource to the research community. The only major concern is that the mice should be made readily available to the community via a common resource such as Jax. Otherwise, the paper does not report any novel findings and the value and impact would be significantly reduced.

We agree these valuable resources should be made available to the research community. We have a proven track record of sharing materials with the non-profit research community and confirm that these SAM mice will be made available to academic and non-profit researchers by MTA. However, due to the double-blind peer review approach, these details do not appear in the blinded manuscript. We apologize for any confusion this may have caused.

The only other major issue is that in Figures 5, S6, and S7, it seems that they are directly activating PCSK9 and LDLR to increase LDL levels, for which they show the time course of serum LDL levels over many weeks, but they don't show serum PCSK9 levels (the actual target). Why is this? Do they have that data (from the same samples) that they can include? It seems important to understanding the relationship of the target gene with downstream effects.

We thank the reviewer for this suggestion. Previously, we were using the RT-qPCR gene expression data to inform on the target gene's behavior. We agree that directly measuring the actual PCSK9 levels is a stronger readout and we now report PCSK9 serum levels in Figure S6f.

Other points to address.

1. In the original review, I suggested the authors should assess the levels of gRNA expression by qRT-PCR (point #11). In response, they provided qPCR for DNA vector genomes, which is not the same thing (Fig S4D). My point was that the cassettes in the arrays could likely be interfering with transcription and changing expression levels. Ideally, this would be assessed to understand the value of the array format. At a minimum, the title for Fig S4D should be changed from "Detection of AAV gRNAs..."

While we understand the reviewer is looking for direct quantification of each gRNA's expression levels, it was not possible to generate target specific assays that are localized to each specific guide and tracr combination. Instead, we have provided quantification of the tracr present in each sample. As expected, gRNA arrays have a lower Ct value associated with tracr assay than single gRNA vectors delivered at the same titer. Since each gRNA in the array is driven by its own U6 promoter and we

have verified there is no rearrangement in the array, we can be confident that there is no expression difference between the U6 promoters. We included the RT-qPCR levels of the ITRs because it is not possible to completely eliminate contaminating AAV genomes. The RNA samples have been DNase treated to degrade gDNA and as much AAV genome as possible, but it is known that AAV genomes are resistant to DNase degradation (this is why the ITR assay still shows a weak signal). To help alleviate confusion, we have now removed the ITR detection data and updated the title to be “Detection of gRNA Tracr in liver tissues.”

2. “The MCP transcription factors interact with the tracr aptamers to synergistically activate targets.” – what “synergy” is being referred to here?

As described by Konermann et al, the synergy is the pairing of VP64 with additional transcription factor activators to induce a greater response than could be achieved with each factor on its own. To make this clearer, I have adjusted the sentence as follows: “The MCP fused to transcription factors binds the MS2 aptamers to bring VP64, HSF1, and P65 together to synergistically activate targets.”

3. “Thus, the application of SAM activation to an initially unsatisfactory model has enabled the continued study of ATTR pathogenesis in mouse models.” – my understanding is that there is no “pathogenesis” in this model of WT ATTR expression?

That is correct. As described in the text, we sought to model ATTR with an allelic series beginning with the WT human coding sequence. However, the humanized mouse locus failed to express hTTR protein at normal human levels. We made the assumption that modifying the WT human sequence to express the various disease associated point mutants would, at best, have a similarly low expression level. For this reason, we decided to focus on optimizing WT expression before engineering disease variants. The text has been further modified for clarity: Thus, SAM activation was able to sufficiently increase WT hTTR expression in humanized mice as the first step in the generation of an ATTR mouse model.

4. Fig 6d-f – It’s not clear this is necessary. Maybe it should be its own figure since it doesn’t relate to Fig 6a-c.

We agree with the reviewer and have now moved 6d-f to figure 7a-c.

5. Fig S2b – It’s confusing what is being measured here. Are they plotting levels of B2m gene expression on a plot that is labelled “dCas9 SAM mRNA”?

We thank the reviewer for this observation. The *B2m* Ct values should have been removed as it is a housekeeping reference gene.

Reviewer #2 (Remarks to the Author):

1. One remaining concern is statistical analysis: The authors state in the figures “T test” and in the tables “Multiple T Test”. What kind of t-test was used? For example, Student’s t-test or Welch’s t-test? Paired or unpaired? One or two tailed? These need to be clarified. “Multiple T test” is a confusing term. Do the authors mean multiple-group comparisons? If so, appropriate multiple-group comparisons such as one- or

two-way ANOVA should be used. Suggest to also provide appropriate justification for the statistical analysis used.

We agree with the reviewer that “Multiple T test” is a confusing term; however, it is the term used by GraphPad Prism. We can confirm that the “Multiple T test” in Prism refers to an unpaired, 2-tailed Student’s T-Test that is performed on each row. We prefer this type of analysis to track the changes between groups at each timepoint to better understand the progression mediated by each delivery method/dose.

2. Suggest providing molecular mass for Western blots in Figures 1b, 4b, and S2a.

We agree with the reviewer and have now added these details to the figure legends.

3. Figure 2f: It appears the images have different magnification. Suggest to include scale bars.

We agree that some images look to be a different size, but we can confirm that all tissue types were imaged at the same magnification. We did note that some images of the heart did not have as much thymus attached as others and have now corrected this.

Reviewers' Comments:

Reviewer #1:

Remarks to the Author:

My concerns have been adequately addressed. I do not understand the resistance to putting the mice into public repositories such as Jax, where they would only be made available to non-profit entities, but I leave it to editorial discretion as to how to handle that issue.

Reviewer #2:

Remarks to the Author:

No further comments

January 14, 2021

Dear reviewers,

Thank you for your constructive comments to improve the manuscript. We provide below specific responses to the questions and concerns raise by you.

REVIEWERS' COMMENTS

Reviewer #1 (Remarks to the Author):

My concerns have been adequately addressed. I do not understand the resistance to putting the mice into public repositories such as Jax, where they would only be made available to non-profit entities, but I leave it to editorial discretion as to how to handle that issue.

We agree these valuable resources should be made available to the research community. Regeneron Pharmaceuticals is dedicated to sharing materials with the non-profit research community and confirm that these SAM mice will be made available by contacting Regeneron Pharmaceuticals, Inc. (email address: preclinical.collaborations@regeneron.com) for academic and non-profit purposes only under an MTA, which allows the use of mice for academic but not commercial purposes

Reviewer #2 (Remarks to the Author):

No further comments